# Interventional Fairness on Partially Known Causal Graphs: A Constrained Optimization Approach

**Aoqi Zuo[1], Yiqing Li[1,2,3], Susan Wei[1] & Mingming Gong[1,3]**

[1]School of Mathematics and Statistics, The University of Melbourne

[2]School of Data Science, Fudan University

[3]Mohamed bin Zayed University of Artificial Intelligence

`azuo@student.unimelb.edu.au`

`yiqingli20@fudan.edu.cn`

`{susan.wei,mingming.gong}@unimelb.edu.au`

## Abstract

Fair machine learning aims to prevent discrimination against individuals or sub-populations based on sensitive attributes such as gender and race. In recent years, causal inference methods have been increasingly used in fair machine learning to measure unfairness by causal effects. However, current methods assume that the true causal graph is given, which is often not true in real-world applications. To address this limitation, this paper proposes a framework for achieving causal fairness based on the notion of interventions when the true causal graph is partially known. The proposed approach involves modeling fair prediction using a Partially Directed Acyclic Graph (PDAG), specifically, a class of causal DAGs that can be learned from observational data combined with domain knowledge. The PDAG is used to measure causal fairness, and a constrained optimization problem is formulated to balance between fairness and accuracy. Results on both simulated and real-world datasets demonstrate the effectiveness of this method.

## 1 Introduction

Machine learning algorithms have demonstrated remarkable success in automating decision-making processes across a wide range of domains (e.g., hiring decisions (Hoffman et al., 2018), recidivism predictions (Dieterich et al., 2016; Brennan et al., 2009), and finance (Sweeney, 2013; Khandani et al., 2010)), providing valuable insights and predictions. However, it has become increasingly evident that these algorithms are not immune to biases in training data, potentially perpetuating discrimination against individual or sub-population group with respect to *sensitive attributes*, e.g., gender and race. For example, bias against female was found in a recruiting tool built by one of Amazon's AI team to review job applicants' resume in a period of time (Kodiyan, 2019).

To achieve fair machine learning, various methods have been proposed with respect to different fairness measures. These methods can be broadly classified into two groups. The first group focuses on developing statistical fairness measures, which typically indicate the statistical discrepancy between individuals or sub-populations, e.g., statistical parity (Dwork et al., 2012), equalized odds (Hardt et al., 2016), and predictive parity (Chouldechova, 2017). The second group is grounded in the causal inference framework (Pearl et al., 2000), which emphasizes understanding the causal relationships between the sensitive attribute and decision outcomes and treats the presence of causal effect of the sensitive attribute on the decision as discrimination (Zhang et al., 2017; Kilbertus et al., 2017; Kusner et al., 2017; Zhang & Bareinboim, 2018b;a; Nabi & Shpitser, 2018; Wu et al., 2019b; Khademi et al., 2019; Chiappa, 2019; Russell et al., 2017; Zhang et al., 2018; Zhang & Wu, 2017; Kusner et al., 2019; Salimi et al., 2019; Wu et al., 2018; Galhotra et al., 2022; Zuo et al., 2022).

Causality-based fairness notions are defined within the framework of Pearl's ladder of causation, which encompasses interventions and counterfactuals. Build on the highest rung and also the most fine-grained type of inference in the ladder, counterfactual fairness (Kusner et al., 2017; Chiappa,

2019; Russell et al., 2017; Wu et al., 2019a) requires the full knowledge of structural causal model or the computation of counterfactuals in the sense of Pearl et al. (2000), thus posing extra challenges compared to the one based on interventions. As the most basic and general notion of causal fairness that is testable with observational data, the interventional fairness assesses the unfairness as the causal effect of the sensitive attribute on the outcome on paths defined by specific attributes. In this paper, we aim to achieve interventional fairness.

The majority of existing methods for ensuring causal fairness assume the presence of a causal directed acyclic graph (DAG) (Pearl, 2009), which encodes causal relationships between variables. However, in many real-world scenarios, it is very difficult to fully specify the causal DAG due to a limited understanding of the system under study. One potential approach is to use causal discovery methods to deduce the causal DAG from observational data (Spirtes & Glymour, 1991; Colombo et al., 2014; Spirtes et al., 2000a; Chickering, 2002b; Shimizu et al., 2006; Hoyer et al., 2008; Zhang & Hyvärinen, 2009; Peters et al., 2014; Peters & Bühlmann, 2014). However, determining the true underlying causal graph solely based on observational data is challenging without strong assumptions about the data generation process, such as linearity (Shimizu et al., 2006) or additive noise (Hoyer et al., 2008; Peters et al., 2014). In general cases, causal discovery methods may produce a Markov equivalence class of DAGs that capture the same conditional independencies in the data, represented by a completely partially directed acyclic graph (CPDAG) (Spirtes et al., 2000b; Chickering, 2002b; Meek, 1995; Andersson et al., 1997; Chickering, 2002a). Although incorporating additional background knowledge allows for the discernment of more causal directions, resulting in a maximally partially directed acyclic graph (MPDAG) (Meek, 1995), it is still unlikely to obtain a unique causal DAG.

Following this line of inquiry, a natural question arises: Can we learn interventional fairness when we only have partially knowledge of the causal graph, represented by an MPDAG? [1] Inspired by Zuo et al. (2022), one straightforward way to achieve fair predictions for interventional fairness is to identify the definite non-descendants of the sensitive attribute on an MPDAG (see Section 3.1). While this approach guarantees interventional fairness, disregarding the descendants results in a notable decline in performance. Thus, we propose a constrained optimisation problem that balances the inherent competition between accuracy and fairness (see Section 3.2). In our approach, to measure interventional fairness in MPDAGs, we model the prediction as the effect of all the observational variables. Interestingly, this modeling technique gives rise to another causal MPDAG, which allows us to discuss the identification of interventional fairness criterion formally.

In this paper, we assume the absence of selection bias and latent confounders since causal discovery algorithms themselves faces difficulties in such challenging scenarios. These assumptions align with many recent related work (Zhang et al., 2017; Chiappa, 2019; Chikahara et al., 2021; Wu et al., 2019a). However, we relax the assumption of a fully directed causal DAG. Based on these considerations, our main contributions on achieving interventional fairness on MPDAGs are as follows:

- We propose a modeling technique on the predictor, which gives rise to a causal MPDAG on which it is feasible to perform causal inference formally;
- We analyze the identification condition over interventional fairness measure on the MPDAG;
- We develop a framework for achieving interventional fairness on partially known causal graphs, specifically MPDAGs, as a constrained optimization problem.

## 2 BACKGROUND

### 2.1 STRUCTURAL CAUSAL MODEL AND CAUSAL GRAPH

The structural causal model (SCM) (Pearl et al., 2000) provides a framework for representing causal relations among variables. It comprises a triple $(U, V, F)$, wherein $V$ represents observable endogenous variables and $U$ represents unobserved exogenous variables that cannot be caused by any variable in $V$. The set $F$ consists of functions $f_1, ..., f_n$, each associated with a variable $V_i \in V$ describing how $V_i$ depends on its direct causes: $V_i = f_i(pa_i, U_i)$. Here, $pa_i$ denotes the observed

---

[1] As CPDAG is a special case of MPDAG without background knowledge, we deal with MPDAG generally.

direct causes of $V_i$ and $U_i$ is the set of unobserved direct causes of $V_i$. The exogenous $U_i$'s are required to be mutually independent. The equations in $F$ induce a causal graph $\mathcal{D}$ that represents the relationships between variables, typically in the form of a directed acyclic graph (DAG), where the direct causes of $V_i$ correspond to its parent set in the causal graph.

**DAGs, PDAGs and CPDAGs.** A directed acyclic graph (DAG) is characterized by directed edges and the absence of directed cycles. When some edges are undirected, it is referred to as a partially directed graph (PDAG). The structure of a DAG captures a collection of conditional independence relationships through the concept of *d-separation* (Pearl, 1988). In cases where multiple DAGs encode the same conditional independence relationships, they are considered to be Markov equivalent. The *Markov equivalence class* of a DAG $\mathcal{D}$ can be uniquely represented by a completed partially directed acyclic graph (CPDAG) $\mathcal{C}$, often denoted as $[\mathcal{C}]$.

**MPDAGs.** A maximally oriented PDAGs (MPDAG) (Meek, 1995) can be obtained by applying Meek's rules in Meek (1995) to a CPDAG with a background knowledge constraint. To construct an MPDAG $\mathcal{G}$ from a given CPDAG $\mathcal{C}$ and background knowledge $\mathcal{B}$, Algorithm 1 in Perkovic et al. (2017) can be utilized, in which the background knowledge $\mathcal{B}$ is assumed to be the *direct causal information* in the form $X \to Y$, indicating that $X$ is a direct cause of $Y$.[2] The subset of Markov equivalent DAGs that align with the background knowledge $\mathcal{B}$ can be uniquely represented by an MPDAG $\mathcal{G}$, denoted as $[\mathcal{G}]$. Both a DAG and a CPDAG can be viewed as special cases of an MPDAG, where the background knowledge is fully known and unknown, respectively.

**Density.** A density $f$ over $\mathbf{V}$ is *consistent* with a DAG $\mathcal{D} = (\mathbf{V}, \mathbf{E})$ if it can be factorized as $f(\mathbf{v}) = \prod_{V_i \in \mathbf{V}} f(v_i | pa(v_i, \mathcal{D}))$ (Pearl, 2009; Perkovic et al., 2017). A density $f$ of $\mathbf{V}$ is consistent with an MPDAG $\mathcal{G} = (\mathbf{V}, \mathbf{E})$ if $f$ is consistent with a DAG in $[\mathcal{G}]$.

## 2.2 CAUSAL INFERENCE AND INTERVENTIONAL FAIRNESS

The interventions $do(\mathbf{X} = \mathbf{x})$ or the shorthand $do(\mathbf{x})$ represent interventions that force $\mathbf{X}$ to take certain value $\mathbf{x}$. In a SCM, this means the substitution of the structural equation $\mathbf{X} = f_{\mathbf{x}}(pa_{\mathbf{x}}, U_{\mathbf{x}})$ with $\mathbf{X} = \mathbf{x}$. The causal effect of a set of treatments $\mathbf{X}$ on a set of responses $\mathbf{Y}$ can be understood as the post-interventional density of $\mathbf{Y}$ when intervening on $\mathbf{X}$ via the $do$ operator, which can be denoted as $f(\mathbf{Y} = \mathbf{y} | do(\mathbf{X} = \mathbf{x}))$ or $P(\mathbf{y}_{\mathbf{x}})$ Pearl (1995; 2009). In the context of an MPDAG $\mathcal{G}$, such causal effect is identifiable if it is possible to be uniquely computed. For a formal definition and a brief review on the causal effect identification problem on MPDAG, please refer to Appendix H.

Build on the $do$-operator, interventional fairness criterion (Salimi et al., 2019) is defined upon *admissible attribute*. An *admissible attribute* is one through which the causal path from the sensitive attribute to the outcome is still considered fair. For example, in a construction company hiring scenario, the attribute *strength* is considered as an *admissible attribute* as it is a valid criterion for assessing job candidates. Despite being causally affected by the attribute *gender*, the causal path '*gender → strength → hiring decision*' should not be regarded as discriminatory.

Formally, let $\mathbf{A}$, $Y$ and $\mathbf{X}$ represent the sensitive attributes, outcome of interest and other observable attributes, respectively. The prediction of $Y$ is denoted by $\hat{Y}$. We say the prediction $\hat{Y}$ is interventionally fair with respect to the sensitive attributes $\mathbf{A}$ if, for any given set of admissible attributes $\mathbf{X}_{ad} \subseteq \mathbf{X}$, it satisfies the following condition:

**Definition 2.1** (Interventional fairness). *(Salimi et al., 2019) We say the prediction $\hat{Y}$ is interventionally fair with respect to the sensitive attributes $\mathbf{A}$ if the following holds for any $\mathbf{X}_{ad} = \mathbf{x}_{ad}$:*

$$P(\hat{Y} = y | do(\mathbf{A} = \mathbf{a}), do(\mathbf{X}_{ad} = \mathbf{x}_{ad})) = P(\hat{Y} = y | do(\mathbf{A} = \mathbf{a}'), do(\mathbf{X}_{ad} = \mathbf{x}_{ad})),$$

*for all possible values of $y$ and any value that $\mathbf{A}$ and $\mathbf{X}_{ad}$ can take.*

---

[2]It is worth noting that other forms of background knowledge, such as tier orderings, specific model restrictions, or data obtained from previous experiments, can also induce MPDAGs Scheines et al. (1998); Hoyer et al. (2012); Hauser & Bühlmann (2012); Eigenmann et al. (2017); Wang et al. (2017); Rothenhäusler et al. (2018).

## 3 PROBLEM FORMULATION

In this section, we focus on the challenge of attaining interventional fairness in PDAGs, particular MPDAGs that can be learnt from observational data using causal discovery algorithms (Spirtes & Glymour, 1991; Colombo et al., 2014; Chickering, 2002a). We begin by presenting a basic implication of the definition of interventional fairness as a baseline method for achieving fairness. However, to overcome its limitations, we formulate a constrained optimisation problem.

### 3.1 FAIRNESS UNDER MPDAGS AS A GRAPHICAL PROBLEM

A simple but important implication in achieving interventional fairness is the following:

**Lemma 3.1.** *Let $\mathcal{G}$ be the causal graph of the given model $(U, V, F)$. Then $\hat{Y}$ will be interventionally fair if it is a function of the admissible set $\mathbf{X}_{ad}$ and non-descendants of $\mathbf{A}$.*

Zuo et al. (2022) propose a graphical criterion and algorithms for identifying the ancestral relationship between two vertices on an MPDAGs which we review in Appendix D.1. These findings form the basis for learning (exactly) interventionally fair predictions as indicated by Lemma 3.1. However, we claim that by including descendants, the prediction accuracy is possible to be improved. We propose a constrained optimisation problem that balances the inherent competition between accuracy and fairness.

### 3.2 $\epsilon$-APPROXIMATE INTERVENTIONAL FAIRNESS

We first establish an approximation to interventional fairness to address the problem of learning a fair prediction on an MPDAG.

**Definition 3.1** ($\epsilon$-Approximate Interventional Fairness). *A predictor $\hat{Y}$ is $\epsilon$-approximate interventionally fair with respect to the sensitive attribute $A$ if for any value of admissible set $\mathbf{X}_{ad} = \mathbf{x}_{ad}$, we have that:*

$$|P(\hat{Y} = y | do(\mathbf{A} = \mathbf{a}), do(\mathbf{X}_{ad} = \mathbf{x}_{ad})) - P(\hat{Y} = y | do(\mathbf{A} = \mathbf{a}'), do(\mathbf{X}_{ad} = \mathbf{x}_{ad}))| \le \epsilon$$

*for any $\mathbf{a}' \ne \mathbf{a}$.*

**Objective.** Our objective is to train a model $h_\theta$ mapping from a subset of observable variables to $Y$ with parameter $\theta$ so as to accurately predict $Y$ while simultaneously achieving $\epsilon$-approximate interventional fairness. To accomplish this, we minimize the loss function $\ell(\hat{y}, y)$ under the fairness constraint. Given a dataset with $n$ observations $\{\mathbf{A}^{(i)}, \mathbf{X}^{(i)}, Y^{(i)}\}$ for $i = 1, 2, ..., n$, where $\mathbf{X}_{ad}^{(i)} \subset \mathbf{X}^{(i)}$ represents the admissible set. For a specific intervention on $\mathbf{X}_{ad} = \mathbf{x}_{ad}$, the objective can be formulated as follows:

$$\min_\theta \quad \frac{1}{n} \sum_{i=1}^{n} \ell(\hat{y}_i, y_i) + \lambda |P(\hat{Y}_{\mathbf{A} \leftarrow \mathbf{a}, \mathbf{X}_{ad} \leftarrow \mathbf{x}_{ad}}) - P(\hat{Y}_{\mathbf{A} \leftarrow \mathbf{a}', \mathbf{X}_{ad} \leftarrow \mathbf{x}_{ad}})|, \tag{1}$$

where $P(\hat{Y}_{\mathbf{A} \leftarrow \mathbf{a}, \mathbf{X}_{ad} \leftarrow \mathbf{x}_{ad}})$ and $P(\hat{Y}_{\mathbf{A} \leftarrow \mathbf{a}', \mathbf{X}_{ad} \leftarrow \mathbf{x}_{ad}})$ represent the post-interventional distributions of $\hat{Y}$ under $do(\mathbf{A} = \mathbf{a}, \mathbf{X}_{ad} = \mathbf{x}_{ad})$ and $do(\mathbf{A} = \mathbf{a}', \mathbf{X}_{ad} = \mathbf{x}_{ad})$, respectively. The parameter $\lambda$ balances the trade-off between accuracy and fairness. In this paper, we focus on the modelling and identification of the objective function for a specific intervention on $\mathbf{X}_{ad}$. If we consider multiple values of $\mathbf{X}_{ad}$, the fairness term in Equation (1) can be replaced with the average of $|P(\hat{Y}_{\mathbf{A} \leftarrow \mathbf{a}, \mathbf{X}_{ad} \leftarrow \mathbf{x}_{ad}}) - P(\hat{Y}_{\mathbf{A} \leftarrow \mathbf{a}', \mathbf{X}_{ad} \leftarrow \mathbf{x}_{ad}})|$ over different interventions on $\mathbf{X}_{ad}$.

This formulation raises two key challenges: 1) how to design the model $h_\theta$ (for $\hat{Y}$) over an MPDAG; 2) when and how $P(\hat{Y} = y | do(\mathbf{A} = \mathbf{a}, \mathbf{X}_{ad} = \mathbf{x}_{ad}))$ is identifiable using the observational densities within our modeling framework.

## 4 INTERVENTIONAL FAIRNESS UNDER MPDAGS

As mentioned in Section 1, the underlying causal DAG $\mathcal{D}$ is usually unknown for real-world datasets. At most, we can obtain a refined Markov equivalence class of DAGs represented by an MPDAG

$\mathcal{G}$ from the observational data $(\mathbf{X}, \mathbf{A})$ and additional background knowledge. In this section, we model the predictor as a function of all the other observable variables, regardless of whether they are (possible) descendants or non-descendants of the sensitive attribute. Then we show that such modeling technique leads to another MPDAG over $(\mathbf{X}, \mathbf{A}, \hat{Y})$ which facilitates causal inference.

### 4.1 MODELING AND VERIFICATION

**Modeling.** We illustrate our modeling technique with Figure 1. Let $\mathcal{D} = (\mathbf{V}, \mathbf{E})$ in Figure 1a be the underlying causal DAG over the observational variables $\mathbf{X}$ and $\mathbf{A}$, where $\mathbf{V} = \mathbf{X} \cup \mathbf{A}$, and $f$ be the consistent observational density over $\mathbf{V}$, factorized as $f(\mathbf{v}) = \prod_{V_i \in \mathbf{V}} f(v_i | pa(v_i, \mathcal{D}))$. We model the fair predictor $\hat{Y}$ as a function $h_\theta(\mathbf{x}, \mathbf{a})$ of $\mathbf{x}$ and $\mathbf{a}$ with parameter $\theta$. Under Pearl's SCM framework, our modeling technique implies

1. An underlying causal DAG $\mathcal{D}^*$ over $\mathbf{X}$, $\mathbf{A}$ and $\hat{Y}$ which includes additional edges from $V$ to $\hat{Y}$ for any $V \in \mathbf{V}$ compared with the DAG $\mathcal{D}$, as depicted in Figure 1b;

2. The density $f$ over $\mathbf{V} \cup \hat{Y}$, denoted as $f(\mathbf{v}, \hat{y}) = f(\hat{y}|\mathbf{v})f(\mathbf{v})$, is consistent with $\mathcal{D}^*$, where $\hat{y} = h_\theta(\mathbf{x}, \mathbf{a})$;

3. The density $f(\mathbf{v})$ is consistent with any MPDAG $\mathcal{G}$, where $\mathcal{D} \in [\mathcal{G}]$, and the density $f(\mathbf{v}, \hat{y})$ is consistent with any MPDAG $\mathcal{G}^*$, where $\mathcal{D}^* \in [\mathcal{G}^*]$. Figure 1c and Figure 1d are two examples of $\mathcal{G}$ and $\mathcal{G}^*$, respectively.

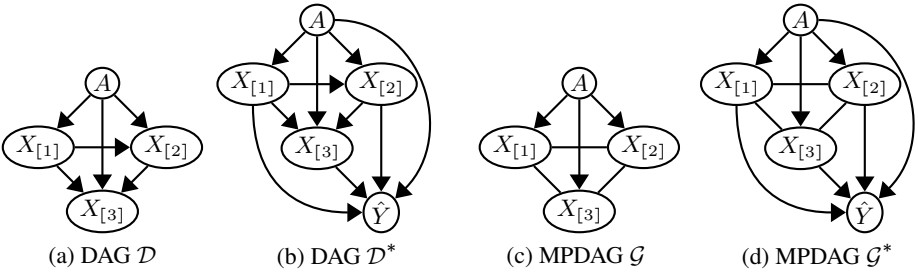

(a) DAG $\mathcal{D}$      (b) DAG $\mathcal{D}^*$      (c) MPDAG $\mathcal{G}$      (d) MPDAG $\mathcal{G}^*$

Figure 1: (a) is an underlying causal DAG $\mathcal{D}$ with three variables $X_{[1]}$, $X_{[2]}$ and $X_{[3]}$ in $\mathbf{X}$; (b) is a causal DAG $\mathcal{D}^*$ under modeling on $\hat{Y}$; (c) is an example MPDAG $\mathcal{G}$ such that $\mathcal{D} \in [\mathcal{G}]$; (d) is an example MPDAG $\mathcal{G}^*$ such that $\mathcal{D}^* \in [\mathcal{G}^*]$.

Next, we introduce Definition 4.1 to formalize such modeling strategy.

**Definition 4.1** (Augmented-$\mathcal{G}$ with $\hat{Y}$). *For a partially directed graph $\mathcal{G} = (\mathbf{V}, \mathbf{E})$, let $\mathcal{G}^*$ augment $\mathcal{G}$ by (i) adding an additional vertex $\hat{Y}$; (ii) adding the edge $V \to \hat{Y}$ for each node $V \in \mathbf{V}$ in $\mathcal{G}$. The resulting graph is denoted as $\mathcal{G}^* = (\mathbf{V}^*, \mathbf{E}^*)$, where $\mathbf{V}^* = \mathbf{V} \cup \hat{Y}$, $\mathbf{E}^* = \mathbf{E} \cup \{V \to \hat{Y} | V \in \mathbf{V}\}$. We call $\mathcal{G}^*$ the augmented-$\mathcal{G}$ with $\hat{Y}$.*

Although directly learning an MPDAG from $\mathbf{X}$, $A$ and $\hat{Y}$ is not feasible due to the unobservability of $\hat{Y}$, Theorem 4.1 implies that once we obtain the MPDAG $\mathcal{G}$ from the observational data $(\mathbf{X}, A)$ with background knowledge $\mathcal{B}$, the augmented-$\mathcal{G}$ with $\hat{Y}$, $\mathcal{G}^*$, is exactly an MPDAG technically such that $\mathcal{D}^* \in [\mathcal{G}^*]$. Therefore, the density $f(\mathbf{v}, \hat{y})$ is consistent with $\mathcal{G}^*$.

**Theorem 4.1.** *For a DAG $\mathcal{D} = (\mathbf{V}, \mathbf{E})$, let $\mathcal{G}$ be an MPDAG consistent with the background knowledge $\mathcal{B}$ such that $\mathcal{D} \in [\mathcal{G}]$. Let $\mathcal{D}^*$ be the augmented-$\mathcal{D}$ with $\hat{Y}$ and $\mathcal{G}^*$ be the augmented-$\mathcal{G}$ with $\hat{Y}$. Then the graph $\mathcal{G}^*$ is an MPDAG consistent with the background knowledge $\mathcal{B} \cup \{V \to \hat{Y} | V \in \mathbf{V}\}$ such that $\mathcal{D}^* \in [\mathcal{G}^*]$.*

Theorem 4.1 is verified by exploring the property of Meek's rules. For more detail, please refer to Appendix B.2. Theorem 4.1 may be of independent interest as it establishes a general modeling method applicable to any problem. The theorem implies that we can directly model $\hat{Y}$ on any MPDAG $\mathcal{G}$ and remarkably, the resulting augmented-$\mathcal{G}$ with $\hat{Y}$ remains an MPDAG consistent with the given background knowledge. This augmented graph enables various causal inference tasks.

### 4.2 IDENTIFICATION OF FAIRNESS CRITERIA

We denote the augmented MPDAG $\mathcal{G} = (\mathbf{V}, \mathbf{E})$ with $\hat{Y}$, where $\mathbf{V} = \mathbf{X} \cup \mathbf{A}$, by $\mathcal{G}^*$. In this section, we discuss the identification of the causal quantity $P(\hat{Y} = y | do(\mathbf{A} = \mathbf{a}), do(\mathbf{X}_{ad} = \mathbf{x}_{ad}))$ in Equation (1) on $\mathcal{G}^*$: when and how we can uniquely express such a causal quantity by the observable densities $f(\mathbf{v})$ over $\mathcal{G}$ and $f(\hat{y} | \mathbf{v})$ over $\mathcal{G}^*$. We start with the general identification problem $P(\hat{Y} = y | do(\mathbf{S} = \mathbf{s}))$, where $\mathbf{S} \subseteq \mathbf{V}$.

Perkovic (2020, Theorem 3.6) establishes the condition and formula for the identification of causal effect $P(\hat{Y} = y | do(\mathbf{S} = \mathbf{s}))$ in an MPDAG $\mathcal{G}^*$ with density $f(\mathbf{v}, \hat{y})$. Here, under our modeling strategy on $\hat{Y}$, we present such graphical condition over the MPDAG $\mathcal{G}$ and represent the identification formula based on $f(\mathbf{v})$ in $\mathcal{G}$ and $f(\hat{y} | \mathbf{v})$ in $\mathcal{G}^*$ in Proposition 4.1. We first provide the notion of *partial causal ordering* as a preliminary.

**Partial Causal ordering Perkovic (2020).** Let $\mathcal{G} = (\mathbf{V}, \mathbf{E})$ be an MPDAG. Since $\mathcal{G}$ may include undirected edges ($-$), it is generally not possible to establish a causal ordering of a node set $\mathbf{V}' \in \mathbf{V}$ in $\mathcal{G}$. Instead, a *partial causal ordering*, $<$, of $\mathbf{V}'$ of $\mathcal{G}$ is defined as a total causal ordering of pairwise disjoint node sets $\mathbf{V_1}, ... \mathbf{V_k}, k \geq 1, \cup_{i=1}^{k} \mathbf{V_i} = \mathbf{V}'$ that satisfies the following condition: if $\mathbf{V_i} < \mathbf{V_j}$ and there is an edge between $V_i \in \mathbf{V_i}$ and $V_j \in \mathbf{V_j}$ in $\mathcal{G}$, then $V_i \rightarrow V_j$ is in $\mathcal{G}$.

Perkovic (2020) develops an algorithm for decomposing a set of nodes as partial causal orderings (PCO) in an MPDAG $\mathcal{G}$. We provide it in Algorithm 2 in Appendix A, which acts as an important component for Proposition 4.1. We denote the parents of the node $W$ in a graph $\mathcal{G}$ as $pa(W, \mathcal{G})$.

**Proposition 4.1.** *Let $\mathbf{S}$ be a node set in an MPDAG $\mathcal{G} = (\mathbf{V}, \mathbf{E})$ and let $\mathbf{V}' = \mathbf{V} \backslash \mathbf{S}$. Furthermore, let $(\mathbf{V_1}, ..., \mathbf{V_k})$ be the output of PCO($\mathbf{V}, \mathcal{G}$). The augmented-$\mathcal{G}$ with $\hat{Y}$ is denoted as $\mathcal{G}^*$. Then for any density $f(\mathbf{v})$ consistent with $\mathcal{G}$ and the conditional density $f(\hat{y} | \mathbf{v})$ entailed by $\mathcal{G}^*$, we have*

  *1. $f(\mathbf{v}, \hat{y}) = f(\hat{y} | \mathbf{v}) \prod_{\mathbf{V_i} \subseteq \mathbf{V}} f(\mathbf{v_i} | pa(\mathbf{v_i}, \mathcal{G}))$,*

  *2. If and only if there is no pair of nodes $V \in \mathbf{V}'$ and $S \in \mathbf{S}$ such that $S - V$ in $\mathcal{G}$, $f(\mathbf{v}' | do(\mathbf{s}))$ and $f(\hat{y} | do(\mathbf{s}))$ are identifiable, and*

$$f(\mathbf{v}' | do(\mathbf{s})) = \prod_{\mathbf{V_i} \subseteq \mathbf{V}'} f(\mathbf{v_i} | pa(\mathbf{v_i}, \mathcal{G})) \tag{2}$$

$$f(\hat{y} | do(\mathbf{s})) = \int f(\hat{y} | \mathbf{v}) f(\mathbf{v}' | do(\mathbf{s})) d\mathbf{v}' = \int f(\hat{y} | \mathbf{v}) \prod_{\mathbf{V_i} \subseteq \mathbf{V}'} f(\mathbf{v_i} | pa(\mathbf{v_i}, \mathcal{G})) d\mathbf{v}' \tag{3}$$

*for values $pa(\mathbf{v_i}, \mathcal{G})$ of $Pa(\mathbf{v_i}, \mathcal{G})$ that are in agreement with $\mathbf{s}$.*

The proof of Proposition 4.1 is based on Perkovic (2020, Theorem 3.6) and Theorem 4.1. The detailed proof is provided in Appendix B.3.

**Example.** We provide a simple example to illustrate Proposition 4.1. Consider the MPDAG $\mathcal{G}$ and $\mathcal{G}^*$ in Figure 1c and Figure 1d, where $\mathcal{G}^*$ is the augmented-$\mathcal{G}$ with $\hat{Y}$. The partial causal ordering of $\mathbf{X} \cup A$ on $\mathcal{G}$ is $\{\{A\}, \{\mathbf{X}\}\}$. $f(\mathbf{x}, a)$ is a density consistent with $\mathcal{G}$, $\mathcal{G}^*$ entails a conditional density $f(\hat{y} | \mathbf{x}, a)$. Then, by Proposition 4.1 we have $f(\mathbf{x}, a, \hat{y}) = f(\hat{y} | \mathbf{x}, a) f(\mathbf{x} | a) f(a)$ in $\mathcal{G}^*$. Since there is no pair of nodes $A$ and $X \in \mathbf{X}$ such that $A - X$ is in $\mathcal{G}$, we have $f(\mathbf{x} | do(a)) = f(\mathbf{x} | a)$ and $f(\hat{y} | do(a)) = \int f(\hat{y} | \mathbf{x}, a) f(\mathbf{x} | a) d\mathbf{x}$ in $\mathcal{G}^*$. For more example, please refer to Appendix C.

**Dealing with non-identification.** In cases where the causal effect of $\mathbf{S}$ on $\hat{Y}$ is not identifiable, we can list MPDAGs corresponding to all valid combinations of edge orientations for edges $V - S$, where $V \in \mathbf{V}'$ and $S \in \mathbf{S}$. In this case, the causal effect in each MPDAG can be identified as a unique functional relationship of the observable density. To address our constrained optimisation problem, we can replace the unfairness term in Equation (1) with the average of unfairness over different MPDAGs. The experimental analysis on unidentifiable cases is provided is Appendix G.11.

**Identification of** $P(\hat{Y} = y | do(\mathbf{A} = \mathbf{a}), do(\mathbf{X}_{ad} = \mathbf{x}_{ad}))$ **in the fairness context.** Under the modeling technique that $\hat{y} = h_\theta(\mathbf{x}, \mathbf{a})$, Proposition 4.1 implies that the causal effect of $\mathbf{A} \cup \mathbf{X}_{ad}$ on $\hat{Y}$ is identifiable for a given set of admissible attributes $\mathbf{X}_{ad} \subset \mathbf{X}$ if and only if there is no

pair of nodes $X \in \mathbf{A} \cup \mathbf{X}_{ad}$ and $V \in \mathbf{V} \backslash (\mathbf{A} \cup \mathbf{X}_{ad})$ such that $X - V$ is in $\mathcal{G}$. In such case, $f(\hat{y}|do(\mathbf{a}), do(\mathbf{x}_{ad})) = \int f(\hat{y}|\mathbf{v}) \prod_{\mathbf{V_i} \subseteq \mathbf{V}} f(\mathbf{v_i}|pa(\mathbf{v_i}, \mathcal{G})) d\mathbf{v}'$ for values $pa(\mathbf{v_i}, \mathcal{G})$ of $Pa(\mathbf{v_i}, \mathcal{G})$ that are in agreement with $\mathbf{a}$ and $\mathbf{x}_{ad}$, where $\mathbf{V}' = \mathbf{V} \backslash (\mathbf{A} \cup \mathbf{X}_{ad})$, $(\mathbf{V_1}, ..., \mathbf{V_m}) = \mathtt{PCO}(\mathbf{V}', \mathcal{G})$.

## 4.3 SOLVING THE OPTIMISATION PROBLEM

Addressing the optimisation problem stated in Equation (1) requires measuring the discrepancy between distributions $P(\hat{Y} = y|do(\mathbf{A} = \mathbf{a}, \mathbf{X}_{ad} = \mathbf{x}_{ad}))$ and $P(\hat{Y} = y|do(\mathbf{A} = \mathbf{a}', \mathbf{X}_{ad} = \mathbf{x}_{ad}))$. Here, we employ Maximum Mean Discrepancy (MMD) (Gretton et al., 2007), but other measures can also be utilized. On the other hand, as evident from the identification formula Equation (3), we need to estimate the stable conditional density $f(\mathbf{v_i}|pa(\mathbf{v_i}, \mathcal{G}))$ and design the model $\hat{y} = h(\mathbf{x}, \mathbf{a})$ to approximate $f(\hat{y}|\mathbf{v})$. However, computing the MMD of two distributions entailed by a model $h$ is intractable. As a result, we resort to Monte Carlo sampling to approximate the integrals involved in the identification formula. For convenience, we estimate $f(\mathbf{v_i}|pa(\mathbf{v_i}, \mathcal{G}))$ using a conditional multivariate normal distribution, but other conditional density estimation approaches can also be employed. Then we generate the interventional data for $\mathbf{v}$ under each intervention $do(\mathbf{A} = \mathbf{a}, \mathbf{X}_{ad} = \mathbf{x}_{ad})$. Finally, a neural net can be trained for $h(\mathbf{x}, \mathbf{a})$. For more on the MMD formulation and its application in our context, please refer to Appendix E.

## 4.4 APPLICABILITY TO OTHER CAUSAL FAIRNESS NOTIONS

Causal fairness notions are defined on different types of causal effects, such as interventional fairness on interventional effects, path-specific fairness on nested counterfactual queries, and counterfactual fairness on counterfactual effects. Their identifiability depends on the identifiability of these causal effects. Our proposed method is directly applicable to other interventional-based fairness notions but not to path-specific causal fairness or counterfactual fairness due to the ongoing challenge of (nested) counterfactual identification over MPDAGs. For more details on the related fairness notions and applicability, please refer to Appendix F.

## 5 EXPERIMENT

In this section, we illustrate our approach on both synthetic and real-world datasets. We measure prediction performance using root mean squared error (RMSE) or accuracy and assess interventional unfairness by MMD. Here, we focus on the scenario where the admissible variable set is empty. Additional experiments involving non-empty admissible variable sets can be found in Appendix G.6.

**Baselines.** We consider three baselines: 1) `Full` model makes predictions using all attributes, including the sensitive attributes, 2) `Unaware` model uses all attributes except the sensitive attributes, and 3) `IFair` is a model mentioned in Section 3.1 which makes predictions using all definite non-descendants of the sensitive attribute in an MPDAG [3]. Our proposed method is $\epsilon$-`IFair`, which uses all attributes and implement the constrained optimization problem in Section 3.2.

## 5.1 SYNTHETIC DATA

We first randomly generate DAGs with $d$ nodes and $s$ directed edges from the graphical model Erdős-Rényi (ER), where $d$ is chosen from $\{5, 10, 20, 30\}$ and the correspongding $s$ is $\{8, 20, 40, 60\}$. For each setting, we generate 10 graphs. For each DAG $\mathcal{D}'$, we consider the last node in topological ordering as the outcome variable and we randomly select one node from the graph as the sensitive attribute. The sensitive attribute can have two or three values, drawn from a Binomial([0,1]) or Multinomial([0,1,2]) distribution separately. The weight, $\beta_{ij}$, of each directed edges $X_i \rightarrow X_j$ in the generated DAG, is drawn from a Uniform($[-1, -0.1] \cup [0.1, 1]$) distribution. The synthetic data is generated according to the following linear structural equation:

$$X_i = \sum_{X_j \in pa(X_i)} \beta_{ij} X_j + \epsilon_i, \tag{4}$$

---

[3] Proposition 4.1 and Theorem D.2 indicate that when the total effect of singleton $A$ on $\hat{Y}$ is identifiable in the augmented MPDAG $\mathcal{G}^*$, the ancestral relationship between $A$ and any other attribute is definite.

where $\epsilon_i$ are independent $N(0, 1)$. Then we generate a sample with size 1000 for each DAG as the observational data. The proportion of training, validation and test data is split as $8 : 1 : 1$.

We consider the DAG $\mathcal{D}$ over all of the variables, excluding the outcome. As the simulated DAG is known, the CPDAG can be obtained from the true DAG without running the causal discovery algorithms. [4] Once we obtain the CPDAG $\mathcal{C}$, where $\mathcal{D} \in [\mathcal{C}]$, we randomly generate the direct causal information $S \rightarrow T$ as the background knowledge from the edges where $S \rightarrow T$ is in DAG $\mathcal{D}$, while $S - T$ is in CPDAG $\mathcal{C}$. Combining with the background knowledge that is necessary to identify fairness, we can obtain the corresponding MPDAG $\mathcal{G}$. We show a randomly generated DAG $\mathcal{D}$, the corresponding CPDAG $\mathcal{C}$ and MPDAG $\mathcal{G}$ as an example in Figure 10, see Appendix G.1.

To measure the unfairness, we generate 1000 interventional data points $\mathbf{X}_{A \leftarrow a}$ under different interventions on $A$ according to the identification formula. To approximate each term $f(\mathbf{v_i}|pa(\mathbf{v_i}, \mathcal{G}))$ in the formula, we fit a conditional multivariate Gaussian distribution using observational data. The generation of $\mathbf{v_i}$ is based on the fitted density and follows the partial causal ordering. The proportion of training and validation for interventional data is split as $8 : 2$. Besides, we generate 1000 interventional data from the ground-truth SCM for different interventions on the sensitive attribute for test. Then we fit all models with the data. The $\lambda$ in our optimisation problem is [0, 0.5, 5, 20, 60, 100]. For additional experiments on nonlinear structural equations and varying amounts of background knowledge, please refer to Appendix G.7 and Appendix G.8, respectively. We also analyze the model robustness experimentally when the graph is learnt by causal discovery algorithms in Appendix G.9. The model robustness on fitting of conditional densities is provide in Appendix G.10.

**Results.** For each graph setting, we report the average unfairness and RMSE achieved on 10 causal graphs in the corresponding trade-off plots in Figure 2. Obviously, both `Full` and `Unaware` methods exhibit lower RMSE but higher unfairness. The `IFair` method achieves nearly 0 unfairness but at the cost of significantly reduced prediction performance. However, our $\epsilon$-`IFair` approach allows for a trade-off between unfairness and RMSE with varying $\lambda$. For certain values of $\lambda$, we can even simultaneously achieve low unfairness comparable to `IFair` model and low RMSE comparable to `Full`. Exemplary density plots of the predictions under two interventional datasets are shown in Figure 3. The degree of overlap in the distributions indicates the fairness of the predictions with respect to the sensitive attribute. The more the distributions overlap, the fairer the predictions are considered to be. More discussion on the accuracy-fairness trade-off is included in Appendix I.

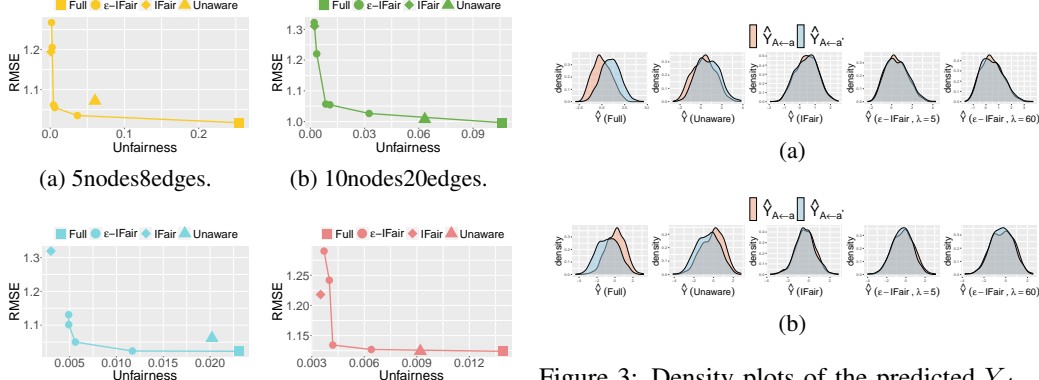

(a) 5nodes8edges.

(b) 10nodes20edges.

(c) 20nodes40edges.

(d) 30nodes60edges.

Figure 2: Accuracy fairness trade-off.

(a)

(b)

Figure 3: Density plots of the predicted $Y_{A \leftarrow a}$ and $Y_{A \leftarrow a'}$ in synthetic data.

## 5.2 REAL DATA

### 5.2.1 THE UCI STUDENT DATASET

Our first experiment with real-world data is based on the UCI Student Performance Data Set (Cortez & Silva, 2008), which contains information about students performance in Mathematics. The dataset consists of records for 395 students, with 32 school-related features. In this dataset, we consider the

---

[4]With an ample sample size, existing causal discovery algorithms have demonstrated high accuracy in recovering the CPDAG from simulated data Glymour et al. (2019).

attribute sex as the sensitive attribute. We create the target attribute Grade as the average of grades in three tests. Our experiments are carried out on the MPDAG $\mathcal{G}$ in Figure 11c. Due to the space limit, Figure 11c is provided in Appendix G.2. For details on graph learning, interventional data generation and model training, please refer to Appendix G.3.

We measure interventional fairness and accuracy using a similar approach as described in Section 5.1. The trade-off plot is shown in Figure 4a. The model `Full` and `Unaware` exhibit higher unfairness. The model `IFair` achieves interventionally fairness at the cost of increased RMSE. On the other hand, our $\epsilon$-`IFair` model strikes a balance between unfairness and RMSE by tuning the values of $\lambda$. Since the test set only contains 19 interventional data for $do(\text{sex} = \text{female})$ and 20 interventional data for $do(\text{sex} = \text{male})$, the unfairness measure may not be highly accurate. Therefore, we also provide the trade-off plot on the training set in Figure 4b, where the unfairness approaches zero for the `IFair` model and $\epsilon$-`IFair` model with stricter penalties on unfairness. Furthermore, the distribution of predictions on the two interventional datasets follows a similar trend as the synthetic data, as depicted in Figure 5.

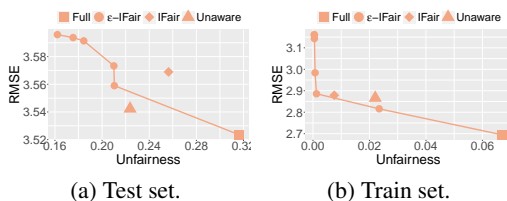

(a) Test set.      (b) Train set.

Figure 4: Accuracy fairness trade-off.

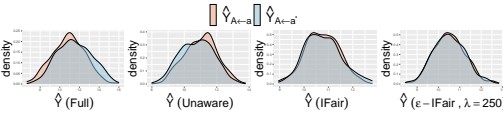

Figure 5: Density plot of the predicted $Y_{A \leftarrow a}$ and $Y_{A \leftarrow a'}$ in Student data.

### 5.2.2 CREDIT RISK DATASET

The task of credit risk assessment involves predicting the likelihood of a borrower defaulting on a loan. For our experiment, we utilize the Credit Risk Dataset, which contains 11 features related to the repayment capability of 32,581 borrowers. In this dataset, we consider the attribute Age as the sensitive attribute and the target variable is Loan_status, which is a binary variable indicating default (1) or no default (0). We focus on two specific age groups: 23 and 30, representing the first and third quantiles, respectively. Our experiments are based on the MPDAG $\mathcal{G}$ in Figure 13a. Due to the space limit, it is provided in Appendix G.4. For details on graph learning, interventional data generation and model training, please refer to Appendix G.5.

Given that the target variable Loan_status is binary, we measure unfairness using the absolute difference in the means of predictions for the interventions on Age = 23 and Age = 30. Prediction performance is evaluated by accuracy. Since there is no non-descendant of Age, we do not have the model `IFair` for this dataset. The trade-off plot is presented in Figure 6a and the distribution of predictions for the two interventional datasets is depicted in Figure 6b. Both of them follow a similar trend as previous.

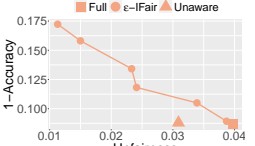

(a) Accuracy fairness trade-off.

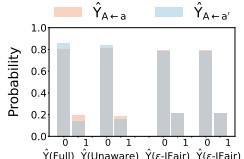

(b) Histogram, where the $\lambda$ for $\epsilon$-`IFair` model is 10 and 150, respectively.

Figure 6: Results on Credit Risk dataset.

## 6 CONCLUSION

This paper presents a framework for achieving interventional fairness on partially known causal graphs, specifically MPDAGs. By leveraging the concept of interventions and modeling fair predictions as the effect of all observational variables, the proposed approach addresses the limitations of existing methods that assume a fully known causal DAG. Through the analysis of identification criteria and the formulation of a constrained optimization problem, the framework provides a principled approach to achieving interventional fairness while maximizing data utility. Experimental results on simulated and real-world datasets demonstrate the effectiveness of the proposed framework. The limitation of this work is that it assumes no selection bias or latent confounders. Future work can focus on extending the proposed framework and addressing these challenges.

## 7 ACKNOWLEDGEMENT

AZ was supported by Melbourne Research Scholarship from the University of Melbourne. This research was undertaken using the LIEF HPC-GPGPU Facility hosted at the University of Melbourne. This Facility was established with the assistance of LIEF Grant LE170100200. SW was supported by ARC DE200101253. MG was supported by ARC DE210101624.

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

# Appendix

## Table of Contents

## A  PRELIMINARIES

**Graph and Subgraphs.** Let $\mathcal{G} = (\mathbf{V}, \mathbf{E})$ denote a graph, where $\mathbf{V} = \{X_1, ..., X_p\}$ represents a set of nodes (variables) and $\mathbf{E}$ represents a set of edges. In our consideration, the graphs can include both directed ($\rightarrow$) and undirected (-) edges, with at most one edge allowed between any two nodes. An *induced subgraph* of $\mathcal{G}$ denoted by $\mathcal{G}_{\mathbf{V}'} = (\mathbf{V}', \mathbf{E}')$ is a subgraph where $\mathbf{V}' \subseteq \mathbf{V}$ and $\mathbf{E}' \subseteq \mathbf{E}$ where $\mathbf{E}'$ consists of all the edges in $\mathbf{E}$ that connects nodes in $\mathbf{V}'$.

**Path.** In the graph $\mathcal{G}$, a path $p$ from $S$ to $T$ is defined as a sequence of distinct nodes $\langle S, ..., T \rangle$ where each consecutive pair of nodes is connected by an edge. Let $p = \langle S = V_0, ..., V_k = T \rangle$ represents a path in a graph $\mathcal{G}$, we say $p$ is a *causal path* from $S$ to $T$ if the direction of the edges follows a pattern $V_i \rightarrow V_{i+1}$ for all $0 \le i \le k - 1$. On the other hand, if no edge $V_i \leftarrow V_{i+1}$ exists in $\mathcal{G}$, then $p$

is a *possibly causal path* from $S$ to $T$. If there exists an edge $V_i \leftarrow V_{i+1}$ in $\mathcal{G}$, $p$ is considered a *non-causal path* in $\mathcal{G}$. A cycle in the graph can be categorized as a causal cycle, possibly causal cycle, or non-causal cycle, depending on whether it represents a causal, possibly causal, or non-causal path from a vertex to itself. A path from $\mathbf{S}$ to $\mathbf{T}$ is said to be *proper* with respect to $\mathbf{S}$ if only its first node is a member of $\mathbf{S}$.

**Colliders, Skeleton and Pattern.** If, in a path $p = \langle X = V_0, ..., V_k = Y \rangle$, there exists a node $V_i$ such that $V_{i-1} \rightarrow V_i$ and $V_i \leftarrow V_{i+1}$, then we say $V_i$ is a *collider* on the path $p$. If $V_{i-1}$ and $V_{i+1}$ are not adjacent, then the triple $\langle V_{i-1}, V_i, V_{i+1} \rangle$ is called a *v-structure collided* on $V_i$, and $V_i$ can be called *unshielded collider*. For a graph $\mathcal{G}$, the *skeleton* of $\mathcal{G}$ is the undirected graph obtained by replacing the directed edges in $\mathcal{G}$ with undirected edges. The *skeleton of $\mathcal{G}$ with unshielded colliders* is the skeleton of $\mathcal{G}$ with the edges resulting unshielded colliders directed. [5]

**MPDAGs Construction.** Algorithm 1 mentioned in (Perkovic et al., 2017) summarizes, the process of constructing the maximal PDAG $\mathcal{G}'$ from the given maximal PDAG $\mathcal{G}$ and backgroud knowledge $\mathcal{B}$. This construction utilizes Meek's rule shown in Figure 7. Specifically, the background knowledge $\mathcal{B}$ is assumed to be the *direct causal information* in the form $S \rightarrow T$, indicating that $S$ is a direct cause of $T$. If Algorithm 1 does not return FAIL, then it implies that the background knowledge $\mathcal{B}$ and returned maximal PDAG $\mathcal{G}'$ are *consistent* with the input maximal PDAG $\mathcal{G}$.

---

**Algorithm 1** Construct MPDAG (Perkovic et al., 2017; Meek, 1995)

1: **Inputs:** MPDAG $\mathcal{G}$ and Background knowledge $\mathcal{B}$.
2: **Output:** MPDAG $\mathcal{G}'$ or FAIL.
3: Let $\mathcal{G}' = \mathcal{G}$;
4: **while** $\mathcal{B} \neq \varnothing$ **do**
5:      Select an edge $\{S \rightarrow T\}$ in $\mathcal{B}$;
6:      $\mathcal{B} = \mathcal{B} \backslash \{S \rightarrow T\}$;
7:      **if** $\{S - T\}$ OR $\{S \rightarrow T\}$ is in $\mathcal{G}'$ **then**
8:          Orient $\{S \rightarrow T\}$ in $\mathcal{G}'$;
9:          Orienting edges in $\mathcal{G}'$ following the rules in Figure 7 until no edge can be oriented;
10:      **else**
11:          FAIL;

---

Figure 7: Meek's orientation rules: R1, R2, R3 and R4 (Meek, 1995). In each rule, if the graph on the left-hand side is an induced subgraph of a PDAG $\mathcal{G}$, the undirected edge in that subgraph should be oriented according to the direction indicated on the right-hand side.

**Undirected Connected Set.** In a graph $\mathcal{G}$, a set of nodes $\mathbf{S}$ is considered an *undirected connected set*, if for any two distinct nodes $S_i$ and $S_j$ in $\mathbf{S}$, there exists an undirected path from $S_i$ to $S_j$ in $\mathcal{G}$.

**Bucket.** Let $\mathbf{D}$ be a node set in an MPDAG $\mathcal{G} = (\mathbf{V}, \mathbf{E})$. If $\mathbf{B}$ is a maximal *undirected connected subset* of $\mathbf{D}$ in $\mathcal{G}$, we call $\mathbf{B}$ a *bucket* in $\mathbf{D}$.

### A.1 EXISTING RESULTS

**Lemma A.1.** *(Perkovic et al., 2017, Lemma B.1) Let $p = \langle V_1, ..., V_k \rangle$ be a b-possibly causal definite status path in an MPDAG $\mathcal{G}$. If there is a node $i \in \{1, ..., n-1\}$ such that $V_i \rightarrow V_{i+1}$, then $p(V_i, V_k)$ is a causal path in $\mathcal{G}$.*

---

[5]In some paper, the skeleton of a DAG $\mathcal{D}$ with unshielded colliders is also called the *pattern* of $\mathcal{D}$.

**Lemma A.2.** *(Perkovic et al., 2017, Lemma 3.6) Let $S$ and $T$ be distinct nodes in an MPDAG $\mathcal{G}$. If $p$ is a b-possibly causal path from $S$ to $T$ in $\mathcal{G}$, then a subsequence $p^*$ of $p$ forms a b-possibly causal unshielded path from $S$ to $T$ in $\mathcal{G}$.*

**Corollary A.1** (Bucket Decomposition). *(Perkovic, 2020, Corollary 3.4) Let $\mathbf{D}$ be a node set in an MPDAG $\mathcal{G} = (\mathbf{V}, \mathbf{E})$. Then there is a unique partition of $\mathbf{D}$ into $\mathbf{B_1}, ..., \mathbf{B_k}$, $k \geq 1$ in $\mathcal{G}$, where $\mathbf{B_1}, ..., \mathbf{B_k}$ are buckets in $\mathcal{G}$. That is*

- $\mathbf{D} = \bigcup_{i=1}^{k} \mathbf{B_i}$, *and*

- $\mathbf{B_i} \cap \mathbf{B_j} = \varnothing$, $i, j \in \{1, ..., k\}$, $i \neq j$, *and*

- $\mathbf{B_i}$ *is a bucket in $\mathbf{D}$ for each $i \in \{1, ..., k\}$.*

**Lemma A.3.** *(Perkovic, 2020, Lemma 3.5) Let $\mathbf{D}$ be a node set in an MPDAG $\mathcal{G} = (\mathbf{V}, \mathbf{E})$ and let $(\mathbf{B_1}, ..., \mathbf{B_k})$, $k \geq 1$, be the output of $PCO(\mathbf{D}, \mathcal{G})$ which is presented in Algorithm 2. Then for each $i, j \in \{1, ..., k\}$, $\mathbf{B_i}$ and $\mathbf{B_j}$ are buckets in $\mathbf{D}$ and if $i < j$, then $\mathbf{B_i} < \mathbf{B_j}$ in $\mathcal{G}$.*

---

**Algorithm 2** Partial causal ordering (PCO) (Perkovic, 2020, Algorithm 1)

---

1: **Inputs:** Node set $\mathbf{D}$ in MPDAG $\mathcal{G} = (\mathbf{V}, \mathbf{E})$.
2: **Output:** An ordered list $\mathbf{B} = (\mathbf{B_1}, ..., \mathbf{B_k})$, $k \geq 1$ of the bucket decomposition of $\mathbf{D}$ in $\mathcal{G}$.
3: Let $\mathbf{ConComp}$ be the bucket decomposition of $\mathbf{V}$ in $\mathcal{G}$;
4: Let $\mathbf{ConComp}$ be an empty list;
5: **while** $\mathcal{B} \neq \varnothing$ **do**
6:     Let $\mathbf{C} \in \mathbf{ConComp}$;
7:     Let $\bar{\mathbf{C}}$ be the set of nodes in $\mathbf{ConComp}$ that are not in $\mathbf{C}$;
8:     **if** all edges between $\mathbf{C}$ and $\bar{\mathbf{C}}$ are into $\mathbf{C}$ in $\mathcal{G}$ **then**
9:         Remove $\mathbf{C}$ from $\mathbf{ConComp}$;
10:         Let $\mathbf{B_*} = \mathbf{C} \cup \mathbf{D}$;
11:         **if** $\mathbf{B_*} \neq \varnothing$ **then**
12:             Add $\mathbf{B_*}$ to the beginning of $\mathbf{B}$;

---

Perkovic (2020) introduces a necessary criterion, referred to as amenability by (Perković et al., 2015; Perkovic et al., 2017), for the identifiability of causal effects in MPDAGs. This criterion is summarized in Proposition A.1.

**Proposition A.1.** *(Perkovic, 2020, Proposition 3.2) Let $\mathbf{X}$ and $\mathbf{Y}$ be disjoint node sets in an MPDAG $\mathcal{G} = (\mathbf{V}, \mathbf{E})$. If there is a proper possibly causal path from $\mathbf{X}$ to $\mathbf{Y}$ that starts with an undirected edge in $\mathcal{G}$, then the causal effect of $\mathbf{X}$ on $\mathbf{Y}$ is not identifiable in $\mathcal{G}$.*

In Perkovic (2020), it is shown that the condition stated in Proposition A.1 is not only necessary, but also sufficient for the identification of causal effects in MPDAGs. This result is established in Theorem A.1.

**Theorem A.1.** *(Perkovic, 2020, Theorem 3.6) Let $\mathbf{X}$ and $\mathbf{Y}$ be disjoint node sets in an MPDAG $\mathcal{G} = (\mathbf{V}, \mathbf{E})$. If there is no proper possibly causal path from $\mathbf{X}$ to $\mathbf{Y}$ in $\mathcal{G}$ that starts with an undirected edge, then for any density $f$ consistent with $\mathcal{G}$ we have*

$$f(\mathbf{y}|do(\mathbf{x})) = \int \prod_{i=1}^{k} f(\mathbf{b_i}|pa(b_i, \mathcal{G}))d\mathbf{b}$$

*for values $pa(\mathbf{b_i}, \mathcal{G})$ of $Pa(\mathbf{b_i}, \mathcal{G})$ that are in agreement with $\mathbf{x}$, where $(\mathbf{B_1}, ..., \mathbf{B_k}) = PCO(An(\mathbf{Y}, \mathcal{G}_{\mathbf{V}\backslash\mathbf{X}}), \mathcal{G})$ and $\mathbf{B} = An(\mathbf{Y}, \mathcal{G}_{\mathbf{V}\backslash\mathbf{X}})\backslash\mathbf{Y}$.*

**Corollary A.2** (Factorization and truncated factorization formula in MPDAGs). *(Perkovic, 2020, Corollary 3.7) Let $\mathbf{X}$ be a node set in an MPDAG $\mathcal{G} = (\mathbf{V}, \mathbf{E})$ and let $\mathbf{V}' = \mathbf{V}\backslash\mathbf{X}$. Furthermore, let $(\mathbf{V_1}, ..., \mathbf{V_k})$ be the output of $PCO(\mathbf{V}, \mathcal{G})$. Then for any density $f$ consistent with $\mathcal{G}$ we have*

1. $f(\mathbf{v}) = \prod_{\mathbf{V_i} \subseteq \mathbf{V}} f(\mathbf{v_i}|pa(\mathbf{v_i}, \mathcal{G}))$,

2. *If there is no pair of nodes $V \in \mathbf{V}'$ and $X \in \mathbf{X}$ such that $X - V$ is in $\mathcal{G}$, then*

$$f(\mathbf{v}'|do(\mathbf{x})) = \prod_{\mathbf{V_i} \subseteq \mathbf{V}'} f(\mathbf{v_i}|pa(\mathbf{v_i}, \mathcal{G}))d\mathbf{v}'$$

*for values $pa(\mathbf{v_i}, \mathcal{G})$ of $Pa(\mathbf{v_i}, \mathcal{G})$ that are in agreement with* $\mathbf{x}$.

**Property 1.** (Katz et al., 2019, Property 1) If a node $v$ is involved in any of the four Meek rules and if the node $v$ does not have an outgoing edge in the original causal DAG, then the oriented edge (in the right hand side of any of the four rules in Figure 7) is incident to $v$.

**Theorem A.2** (Orientation completeness). *(Meek, 1995, Theorem 3) The result of applying rules R1, R2 and R3 in Figure 7 to a pattern of a DAG is a CPDAG.*

**Theorem A.3** (Orientation completeness w.r.t background knowledge). *(Meek, 1995, Theorem 4) Let $\mathcal{B}$ be a set of background knowledge consistent with a pattern of a DAG. The result of applying rules R1, R2, R3 and R4 (and orienting edges according to $\mathcal{B}$) to the pattern is an MPDAG with respect to $\mathcal{B}$.*

# B    DETAILED PROOFS

## B.1    PROOF OF LEMMA 3.1

*Proof.* Let $W$ be any non-descendant of $A$ in $\mathcal{G}$. Then $P(W|do(A = a)) = P(W|do(A = a')) = P(W)$ (according to the Rule 3 of the do-calculus (Pearl, 2009)). Since $W_{A \leftarrow a}$ and $W_{A \leftarrow a'}$ have the same distribution, for any sensitive attribute, the distribution of $\hat{Y}$ as a function of the non-descendants of $A$ and the intervened admissible variable $\mathbf{X}_{ad}$ is invariant.    □

## B.2    PROOF OF THEOREM 4.1

According to the definition of MPDAG, $\mathcal{G}$ can be induced from the Markov equivalence class of DAG $\mathcal{D} = (\mathbf{V}, \mathbf{E})$ with background knowledge $\mathcal{B}$. By exploring Meek's rules, specifically, Property 1., we show that $\mathcal{G}^*$ is an MPDAG that can be derived from the DAG $\mathcal{D}^*$ with background knowledge $\mathcal{B} \cup \{V \rightarrow \hat{Y}|V \in \mathbf{V}\}$. We provide the details in checking with Meek's rules in this section.

### B.2.1    TECHNICAL LEMMAS

In this section, we introduce some lemmas that are useful in the proof of Theorem 4.1.

**Lemma B.1.** *For a DAG $\mathcal{D} = (\mathbf{V}, \mathbf{E})$, let $\mathcal{D}^*$ be the augmented-$\mathcal{D}$ with $\hat{Y}$. The CPDAG of $\mathcal{D}$ and $\mathcal{D}^*$ are denoted as $\mathcal{C}$ and $\mathcal{C}^*$, respectively. Then, compared with $\mathcal{C}$, newly directed edges in $\mathcal{C}^*$ only contain $V \rightarrow \hat{Y}$ for some $V \in \mathbf{V}$.*

*Proof.* As discussed in Meek Meek (1995), deducing the CPDAG from a DAG consists of two phases:

- Phase I. Find the skeleton of the DAG with unshielded colliders.

- Phase II. On the returned graph by phase I, orient every edge that can be oriented by successive applications of Meek's rules R1, R2 and R3.

So here, before identifying the difference between two CPDAGs $\mathcal{C}$ and $\mathcal{C}^*$, we first identify the difference between two skeletons with unshielded colliders $\mathcal{K}$ and $\mathcal{K}^*$, where $\mathcal{K}$ and $\mathcal{K}^*$ are the corresponding skeletons of the DAG $\mathcal{D}$ and $\mathcal{D}^*$ with unshielded colliders, respectively.

Phase I. Compared with $\mathcal{D}$, the new v-structures in $\mathcal{D}^*$ are edges in $\{V \rightarrow \hat{Y}|V \in \mathbf{V}\}$ that form unshielded colliders on $\hat{Y}$. Therefore, it is obvious that compared with $\mathcal{K}$, the newly directed edges in $\mathcal{K}^*$ are edges in $\{V \rightarrow \hat{Y}|V \in \mathbf{V}\}$ that form unshielded colliders on $\hat{Y}$ in $\mathcal{D}^*$.

Phase II. Suppose the application of Meek's rules to $\mathcal{K}$ results the CPDAG $\mathcal{C}$. Next, we apply Meek's rules on $\mathcal{K}^*$. Since the application of Meek's rules checks one edge per time, orienting edges on $\mathcal{K}^*$ is equivalent to the following recursive two-step procedure:

S1.  Check and orient edges that can be oriented between any $V \in \mathbf{V}$;

S2. Check and orient edges that can be oriented between any $V \in \mathbf{V}$ and $\hat{Y}$ on the returned partially directed graph from the step S1;

S3. Go to S1 until no more edges can be oriented.

Clearly, compared with the CPDAG $\mathcal{C}$ oriented from $\mathcal{K}$, newly directed edges in $\mathcal{C}^*$ only contain $V \to \hat{Y}$ for some $V \in \mathbf{V}$ if and only if any directed edge in $\mathcal{C}$ gets directed in $\mathcal{C}^*$ and the step S1, S2 and S3 in the above does not orient more edges except $V \to \hat{Y}$ for some $V \in \mathbf{V}$ compared with $\mathcal{C}$.

We first prove that all directed edges in $\mathcal{C}$ get directed in $\mathcal{C}^*$ by step S1 and step S1 cannot orient more edges except $V \to \hat{Y}$ for some $V \in \mathbf{V}$ compared with $\mathcal{C}$. Suppose that the CPDAG $\mathcal{C}$ is obtained by applying Meek's rules on $\mathcal{K}$ in the sequence $seq$. We denote the partially directed acyclic graph obtained by applying Meek's rules in the same sequence $seq$ on $\mathcal{K}^*$ as $\mathcal{C}^*_I$. Since the induced subgraph of $\mathcal{K}^*$ over $\mathbf{V}$ is exactly the graph $\mathcal{K}$, therefore, the induced subgraph of $\mathcal{K}^*$ over $\mathbf{V}$ will be oriented exactly as the CPDAG $\mathcal{C}$. Then we check the possibility of the further orientations on edges between any $V \in \mathbf{V}$ considering the edge directions between any $V \in \mathbf{V}$ and $\hat{Y}$. Property 1. implies that no more edges between any $V \in \mathbf{V}$ can be oriented. The returned partially directed acyclic graph from this step is $\mathcal{C}^*_I$. Property 1. also implies that step S2 does not orient more edges except $V \to \hat{Y}$ for some $V \in \mathbf{V}$ compared with $\mathcal{C}$. The returned partially directed acyclic graph from step S2 is denoted as $\mathcal{C}^*_{II}$. Then we move onto step S3 and show that no more edge can be oriented in $\mathcal{C}^*_{II}$ any more. Due to the same reason as in step S1 that there is no edge directed out of $\hat{Y}$ in $\mathcal{D}^*$, no more edges between any $V \in \mathbf{V}$ can be oriented. Therefore, the application of Meek's rules on $\mathcal{K}^*$ stops here. According to the orientation completeness presented in Theorem A.2, the returned graph $\mathcal{C}^*_{II}$ is exactly the CPDAG $\mathcal{C}^*$. $\qquad\square$

**Lemma B.2.** *For a DAG $\mathcal{D} = (\mathbf{V}, \mathbf{E})$, let $\mathcal{D}^*$ be the augmented-$\mathcal{D}$ with $\hat{Y}$, $\mathcal{C}$ and $\mathcal{C}^*$ be the CPDAG of $\mathcal{D}$ and $\mathcal{D}^*$, respectively. Given additional background knowledge $\mathcal{B}_\mathcal{C}$ that is consistent with $\mathcal{C}$, we denote the returned MPDAG as $\mathcal{G}$ when applying Algorithm 1 on $\mathcal{C}$ and $\mathcal{B}_\mathcal{C}$ and as $\mathcal{G}^*$ when applying Algorithm 1 on $\mathcal{C}^*$ and $\mathcal{B}_\mathcal{C}$. Then, compared with $\mathcal{G}$, newly directed edges in $\mathcal{G}^*$ only contain $V \to \hat{Y}$ for some $V \in \mathbf{V}$.*

*Proof.* The proof is similar to the proof of Lemma B.1. According to Lemma B.1, compared with $\mathcal{C}$, the newly directed edges in $\mathcal{C}^*$ only contain $V \to \hat{Y}$ for some $V \in \mathbf{V}$. The application of Algorithm 1 to $\mathcal{C}$ and $\mathcal{B}_\mathcal{C}$ results the MPDAG $\mathcal{G}$. Next, we apply Algorithm 1 on $\mathcal{C}^*$ and $\mathcal{B}_\mathcal{C}$. Since the application of Meek's rules checks one edge per time, orienting edges on $\mathcal{C}^*$ is equivalent to the following recursive two-step procedure:

S1. Check and orient edges that can be oriented between any $V \in \mathbf{V}$;

S2. Check and orient edges that can be oriented between any $V \in \mathbf{V}$ and $\hat{Y}$ on the returned partially directed graph from the last step;

S3. Go to S1 until no more edges can be oriented.

Clearly, compared with the MPDAG $\mathcal{G}$ oriented from $\mathcal{C}$ given the background knowledge $\mathcal{B}_\mathcal{C}$, newly directed edges in $\mathcal{G}^*$ only contain $V \to \hat{Y}$ for some $V \in \mathbf{V}$ if and only if any directed edge in $\mathcal{G}$ gets directed in $\mathcal{G}^*$ and the step S1, S2 and S3 in the above does not orient more edges except $V \to \hat{Y}$ for some $V \in \mathbf{V}$ compared with $\mathcal{G}$.

We first prove that all directed edges in $\mathcal{G}$ get directed in $\mathcal{G}^*$ by step S1 and the step S1 cannot orient more edges except $V \to \hat{Y}$ for some $V \in \mathbf{V}$ compared with $\mathcal{G}$. According to Lemma B.1, the induced subgraph of $\mathcal{C}^*$ over $\mathbf{V}$ is exactly the same as the graph $\mathcal{C}$. Suppose that the MPDAG $\mathcal{G}$ is obtained by applying Meek's rules on $\mathcal{C}$ with the sequence $seq$. We denote the partially directed acyclic graph obtained by applying Meek's rules with the same sequence $seq$ on $\mathcal{C}^*$ as $\mathcal{G}^*_I$. Since the induced subgraph of $\mathcal{C}^*$ over $\mathbf{V}$ is exactly the graph $\mathcal{C}$, therefore, the induced subgraph of $\mathcal{C}^*$ over $\mathbf{V}$ will be oriented exactly as the MPDAG $\mathcal{G}$. Then we check the possibility of the further orientation on edges between any $V \in \mathbf{V}$ with considering the directions between any $V \in \mathbf{V}$ and $\hat{Y}$. Property 1. implies that no more edges between any $V \in \mathbf{V}$ can be oriented. The returned

partially directed acyclic graph from this step is $\mathcal{G}^*{}_I$. Property 1. also implies that step S2 does not orient more edges except $V \to \hat{Y}$ for some $V \in \mathbf{V}$ compared with $\mathcal{G}$. The returned partially directed acyclic graph from step S2 is denoted as $\mathcal{G}^*{}_{II}$. Then we move onto step S3 and show that no more edge can be oriented in $\mathcal{G}^*{}_{II}$ any more. Due to the same reason as in step S1 that there is no edge directed out of $\hat{Y}$ in $\mathcal{D}^*$, no more edges between any $V \in \mathbf{V}$ can be oriented. Therefore, the application of Meek's rules on $\mathcal{C}^*$ stops here. According to the orientation completeness presented in Theorem A.3, the returned graph $\mathcal{G}^*{}_{II}$ is exactly the MPDAG $\mathcal{G}^*$. □

**Lemma B.3.** *For a DAG $\mathcal{D} = (\mathbf{V}, \mathbf{E})$, let $\mathcal{D}^*$ be the augmented-$\mathcal{D}$ with $\hat{Y}$, $\mathcal{C}$ and $\mathcal{C}^*$ be the CPDAG of $\mathcal{D}$ and $\mathcal{D}^*$, respectively. Given additional background knowledge $\mathcal{B}_{\mathcal{C}}$ that is consistent with $\mathcal{C}$, we denote the returned MPDAG as $\mathcal{G}$ when applying Algorithm 1 on $\mathcal{C}$ and $\mathcal{B}_{\mathcal{C}}$. Given the additional background knowledge $\mathcal{B}_{\mathcal{C}^*} = \mathcal{B}_{\mathcal{C}} \cup \{V \to \hat{Y} | V \in \mathbf{V}\}$ that is consistent with $\mathcal{C}^*$, the corresponding MPDAG is denoted as $\mathcal{G}^*$. Then, the MPDAG $\mathcal{G}^*$ is exactly the augmented-$\mathcal{G}$.*

*Proof.* Algorithm 1 can be used to construct the MPDAG from the CPDAG and background knowledge, by leveraging Meek's rules. Since Algorithm 1 checks one background knowledge per time, applying Meek's rules to $\mathcal{C}^*$ and $\mathcal{B}_{\mathcal{C}^*}$ is equivalent to the following two-steps procedures:

   S1. Select the background knowledge in $\mathcal{B}_{\mathcal{C}}$ and check and orient every edge that can be oriented in $\mathcal{C}^*$. The returned partially directed graph is denoted as $\mathcal{G}^*{}_I$.

   S2. Select the background knowledge in $\{V \to \hat{Y} | V \in \mathbf{V}\}$ and check and orient every edge that can be oriented on $\mathcal{G}^*{}_I$.

After the step S1, according to Lemma B.2, compared with $\mathcal{G}$, newly directed edges in $\mathcal{G}^*{}_I$ only contain $V \to \hat{Y}$ for some $V \in \mathbf{V}$. Then we will show that in the step S2, after orienting $V \to \hat{Y}$ for any $V \in \mathbf{V}$ in $\mathcal{C}^*{}_I$, no more edges in $V - V'$ for any $V, V' \in \mathbf{V}$ can be oriented.

According to Property 1., since there is no such edge directed out of $\hat{Y}$ in $\mathcal{D}^*$, no more edge between any $V \in \mathbf{V}$ can be oriented. According to the orientation completeness presented in Theorem A.3, the returned graph from this step is the MPDAG $\mathcal{G}^*$, which is exactly the augmented-$\mathcal{G}$ defined by Definition 4.1. □

### B.2.2 Proof of Theorem 4.1

*Proof.* Figure 8 shows how all lemmas fit together to prove the Theorem 4.1. Suppose the MPDAG $\mathcal{G}$ can be constructed from the Markov equivalence class of the DAG $\mathcal{D} = (\mathbf{V}, \mathbf{E})$ with the background knowledge $\mathcal{B}$. Denote the augmented-$\mathcal{D}$ with $\hat{Y}$ by $\mathcal{D}^*$ and the augmented-$\mathcal{G}$ with $\hat{Y}$ by $\mathcal{G}^*$. Following from Lemma B.1, Lemma B.2 and Lemma B.3, we can see that $\mathcal{G}^*$ can be constructed from the Markov equivalence class of the $\mathcal{D}^*$ with the background knowledge $\mathcal{B} \cup \{V \to \hat{Y} | V \in \mathbf{V}\}$ by leveraging Meek's rules. Therefore, the graph $\mathcal{G}^*$ is still an MPDAG and $\mathcal{D}^* \in [\mathcal{G}^*]$. □

Lemma B.1 $\longrightarrow$ Lemma B.2 $\longrightarrow$ Lemma B.3 $\longrightarrow$ **Theorem 4.1**

Figure 8: Proof structure of Theorem 4.1

### B.3 Proof of Proposition 4.1

We first show in Proposition B.1 a sufficient and necessary condition and the formula for the identification of $P(\hat{Y} = y | do(\mathbf{S} = \mathbf{s}))$ in an MPDAG $\mathcal{G}^*$ based on the consistent density $f(\mathbf{v}, \hat{y})$. Then, utilizing the relationship between the MPDAG $\mathcal{G}$ and $\mathcal{G}^*$, we derive Proposition 4.1.

### B.3.1 Technical lemmas

**Proposition B.1.** *Let $\mathbf{S}$ be a node set in an MPDAG $\mathcal{G} = (\mathbf{V}, \mathbf{E})$ and let $\mathbf{V}' = \mathbf{V} \backslash \mathbf{S}$. Furthermore, let $(\mathbf{V}_1, ..., \mathbf{V}_k)$ be the output of $PCO(\mathbf{V}, \mathcal{G})$. The augmented-$\mathcal{G}$ with node $\hat{Y}$ is denoted as $\mathcal{G}^*$. Then for any density $f(\mathbf{v}, \hat{y})$ consistent with $\mathcal{G}^*$, we have*

*1.* $f(\mathbf{v}, \hat{y}) = f(\hat{y}|\mathbf{v}) \prod_{\mathbf{V_i} \subseteq \mathbf{V}} f(\mathbf{v_i}|pa(\mathbf{v_i}, \mathcal{G}^*))$,

*2.* *If and only if there is no pair of nodes $V \in \mathbf{V}'$ and $S \in \mathbf{S}$ such that $S - V$ in $\mathcal{G}^*$, $f(\mathbf{v}'|do(\mathbf{s}))$ and $f(\hat{y}|do(\mathbf{s}))$ are identifiable, and*

$$f(\mathbf{v}'|do(\mathbf{s})) = \prod_{\mathbf{V_i} \subseteq \mathbf{V}'} f(\mathbf{v_i}|pa(\mathbf{v_i}, \mathcal{G}^*))$$

$$f(\hat{y}|do(\mathbf{s})) = \int f(\hat{y}|\mathbf{v}) f(\mathbf{v}'|do(\mathbf{s})) d\mathbf{v}' = \int f(\hat{y}|\mathbf{v}) \prod_{\mathbf{V_i} \subseteq \mathbf{V}'} f(\mathbf{v_i}|pa(\mathbf{v_i}, \mathcal{G}^*)) d\mathbf{v}'$$

*for values $pa(\mathbf{v_i}, \mathcal{G}^*)$ of $Pa(\mathbf{v_i}, \mathcal{G}^*)$ that are in agreement with $\mathbf{s}$.*

*Proof.* Since in the MPDAG $\mathcal{G}^*$, there is the directed edge $V \to \hat{Y}$ for any $V \in \mathbf{V}$, the output of PCO$(\mathbf{V} \cup \hat{Y}, \mathcal{G}^*)$ is $(\mathbf{V_1}, ..., \mathbf{V_k}, \hat{Y})$. The parent of $\hat{y}$ is $\mathbf{v}$. The first statement then follows from the first statement in Corollary A.2.

For the second statement, PCO$(An(\mathbf{V}', \mathcal{G}^*_{\mathbf{V}' \cup \hat{y}}), \mathcal{G}^*) = $ PCO$(\mathbf{V}', \mathcal{G}^*)$. We first prove the sufficiency. That there is no pair of nodes $V \in \mathbf{V}'$ and $S \in \mathbf{S}$ such that $S - V$ is in the MPDAG $\mathcal{G}^*$ means $S$ and $V$ cannot be in the same bucket. Therefore, some of the buckets $\mathbf{V_i}, i \in \{1, ..., k\}$ in the bucket decomposition of $\mathbf{V}$ will contain only nodes in $\mathbf{S}$. Hence, obtaining the bucket decomposition of $\mathbf{V}'$ is the same as leaving out buckets $\mathbf{V_i}$ that contain nodes in $\mathbf{S}$ from $\mathbf{V_1}, ..., \mathbf{V_k}$. Following from Theorem A.1 when taking $\mathbf{Y} = \mathbf{V}'$, we have $f(\mathbf{v}'|do(\mathbf{s})) = \prod_{\mathbf{V_i} \subseteq \mathbf{V}'} f(\mathbf{v_i}|pa(\mathbf{v_i}, \mathcal{G}^*))$. Then $f(\hat{y}|do(\mathbf{s}))$ can be identified directly. Next, we show the necessity. According to Proposition A.1, the necessary condition for $f(\mathbf{v}'|do(\mathbf{s}))$ to be identifiable is that there is no pair of nodes $V \in \mathbf{V}'$ and $S \in \mathbf{S}$ such that $S - V$ in $\mathcal{G}^*$. The necessary condition for $f(\hat{\mathbf{y}}|do(\mathbf{s}))$ to be identifiable is that there is no proper possibly causal path from $\mathbf{S}$ to $\hat{Y}$ that starts with an undirected edge in $\mathcal{G}^*$. Since $V \to \hat{Y}$ for any $V \in \mathbf{V}$, this necessary condition is equivalent to the condition that there is no proper possibly causal path from $\mathbf{S}$ to $\mathbf{V}'$ that starts with an undirected edge in $\mathcal{G}^*$. This completes the proof of Proposition 4.1. $\square$

### B.3.2 PROOF OF PROPOSITION 4.1

*Proof.* For any density $f(\mathbf{v})$ consistent with the MPDAG $\mathcal{G}$, there exists a DAG $\mathcal{D} \in [\mathcal{G}]$ such that $f(\mathbf{v})$ is consistent with $\mathcal{D}$. Then the density $f(\mathbf{v}, \hat{y})$ factorized as $f(\mathbf{v}, \hat{y}) = f(\hat{y}|\mathbf{v})f(\mathbf{v})$ is consistent with the DAG augmented-$\mathcal{D}$ with $\hat{Y}$, denoted by $\mathcal{D}^*$. Theorem 4.1 implies that $f(\mathbf{v}, \hat{y})$ is consistent with the MPDAG augmented-$\mathcal{G}$ with $\hat{Y}$, denoted by $\mathcal{G}^*$. Therefore, two statements in Proposition B.1 applies here. Besides, we have $f(\mathbf{v_i}|pa(\mathbf{v_i}, \mathcal{G}^*)) = f(\mathbf{v_i}|pa(\mathbf{v_i}, \mathcal{G}))$ for any $\mathbf{V_i} \subseteq \mathbf{V}'$; if there is no pair of nodes $V \in \mathbf{V}'$ and $S \in \mathbf{S}$ such that $S - V$ is in $\mathcal{G}^*$, there is no pair of nodes $V \in \mathbf{V}'$ and $S \in \mathbf{S}$ such that $S - V$ is in $\mathcal{G}$. Modifying the condition for second statement in Proposition B.1 and replacing $f(\mathbf{v_i}|pa(\mathbf{v_i}, \mathcal{G}^*))$ with $f(\mathbf{v_i}|pa(\mathbf{v_i}, \mathcal{G}))$ leads to Proposition 4.1. $\square$

## C AN ILLUSTRATION EXAMPLE FOR PROPOSITION 4.1

Consider the MPDAG $\mathcal{G} = (\mathbf{V}, \mathbf{E})$ in Figure 9a, where $\mathbf{V} = (A, B, C, D, E, R, L, M, N)$. Figure 9b is the augmented-$\mathcal{G}$ with $\hat{Y}$, denoted as $\mathcal{G}^*$. The partial causal ordering of $\mathbf{V}$ on $\mathcal{G}$ is $\{\{B, C\}, \{A, E\}, \{M, L\}, \{D\}, \{R\}, \{N\}\}$. $f(\mathbf{v})$ is a density consistent with $\mathcal{G}$ and $\mathcal{G}^*$ entails a conditional density $f(\hat{y}|\mathbf{v})$. Then, by Proposition 4.1, we have $f(\mathbf{v}, \hat{y}) = f(\hat{y}|\mathbf{v})f(n|a, m, l, r)f(r|e)f(d|b, e)f(m, l)f(a, e)f(b, c)$ in $\mathcal{G}^*$. Let $\mathbf{S} = \{A, E\}$ and $\mathbf{V}' = \mathbf{V} \backslash \mathbf{S}$. Since there is no pair of nodes $S \in \mathbf{S}$ and $V \in \mathbf{V}$ such that $S - V$ is in $\mathcal{G}$, we have $f(\mathbf{v}'|do(\mathbf{s})) = f(n|a, m, l, r)f(r|e)f(d|b, e)f(m, l)f(b, c)$ and $f(\hat{y}|do(\mathbf{s})) = \int f(\hat{y}|\mathbf{v})f(n|a, m, l, r)f(r|e)f(d|b, e)f(m, l)f(b, c)d\mathbf{v}'$ in $\mathcal{G}^*$.

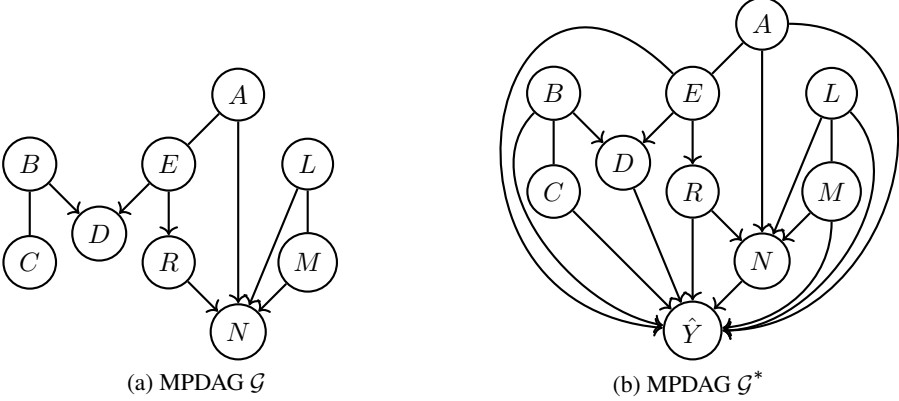

Figure 9: (a) is an MPDAG $\mathcal{G}$; (b) is the extended-$\mathcal{G}$ with $\hat{Y}$, denoted as $\mathcal{G}^*$.

## D    FAIRNESS UNDER MPDAGS AS A GRAPHICAL PROBLEM

In this section, we first review the graphical criterion and algorithms presented in Zuo et al. (2022) for identifying ancestral relations in an MPDAG. Then as a complementary finding to (Zuo et al., 2022, Proposition 5.2), we present a sufficient and necessary graphical criterion that establishes the condition under which any ancestral relationship with regards to a specific vertex in an MPDAG can be definitively determined.

### D.1    IDENTIFYING ANCESTRAL RELATIONS IN AN MPDAG

We first clarify the types of ancestral relations in an MPDAG. With respect to a variable $S$, a variable $T$ can be either

- a *definite descendant* of $S$ if $T$ is a descendant of $S$ in every equivalent DAG,
- a *definite non-descendant* of $S$ if $T$ is a non-descendant of $S$ in every equivalent DAG,
- a *possible descendant* of $S$ if $T$ is neither a definite descendant nor a definite non-descendant of $S$.

Perkovic et al. (2017) state that $T$ is a definite non-descendant of $S$ in an MPDAG $\mathcal{G}$ if and only if there is no possibly causal path from $S$ to $T$ in $\mathcal{G}$. Zuo et al. (2022) propose a sufficient and necessary graphical criterion to identify whether $T$ is a definite descendant of $S$ in an MPDAG in Theorem D.1.

We first introduce some terms as a preliminary. In the graph $\mathcal{G}$, a *chord* of a path refers to any edge that connects two non-consecutive vertices on the path. Conversely, a *chordless path* is a path that does not contain any such edges. A graph is considered *complete* if every pair of distinct vertices in the graph is adjacent to each other. Consider an MPDAG denoted as $\mathcal{G}$. Let $S$ and $T$ represent two distinct vertices in $\mathcal{G}$. The *critical set* of $S$ with respect to $T$ in $\mathcal{G}$ consists of all adjacent vertices of $S$ that lie on at least one chordless, possibly causal path from $S$ to $T$.

**Theorem D.1.** *(Zuo et al., 2022, Theorem 4.5) Let $S$ and $T$ be two distinct vertices in an MPDAG $\mathcal{G}$, and $\mathbf{C}$ be the critical set of $S$ with respect to $T$ in $\mathcal{G}$. Then $T$ is a definite descendant of $S$ if and only if either $S$ has a definite arrow into $\mathbf{C}$, that is $\mathbf{C} \cap ch(S, \mathcal{G}) \neq \varnothing$, or $S$ does not have a definite arrow into $\mathbf{C}$ but $\mathbf{C}$ is non-empty and induces an incomplete subgraph of $\mathcal{G}$.*

Based on Theorem D.1, Zuo et al. (2022) propose Algorithm 3 to identify the type of ancestral relation between two vertices. The set of definite descendants, possible descendants and definite non-descendants of the sensitive attribute can be identified by leveraging Algorithm 3 on each pair of sensitive and any other attribute. In this way, we can select the definite non-descendants to make a counterfactually (and also interventionally) fair prediction or use both definite non-descendants and

possible descendants of $A$ to increase the prediction accuracy at the cost of a violation of counter-factual (and also interventional) fairness.

---

**Algorithm 3** Identify the type of ancestral relation of $S$ with respect to $T$ in an MPDAG (Zuo et al., 2022, Algorithm 2)

---

1: **Input:** MPDAG $\mathcal{G}$, two distinct variables $S$ and $T$ in $\mathcal{G}$.
2: **Output:** The type of ancestral relation between $S$ and $T$.
3: Find the critical set $\mathbf{C}$ of $S$ with respect to $T$ in $\mathcal{G}$ by Algorithm 4.
4: **if** $|\mathbf{C}| = 0$ **then**
5:     **return** $T$ is a definite non-descendant of $S$.
6: **if** $S$ has an arrow into $\mathbf{C}$ or $\mathbf{C}$ induces an incomplete subgraph of $\mathcal{G}$ **then**
7:     **return** $T$ is a definite descendant of $S$.
8: **return** $T$ is a possible descendant of $S$.

---

**Algorithm 4** Finding the critical set of $S$ with respect to $T$ in an MPDAG (Zuo et al., 2022, Algorithm 1)

---

1: **Input:** MPDAG $\mathcal{G}$, two distinct vertices $S$ and $T$ in $\mathcal{G}$.
2: **Output:** The critical set $\mathbf{C}$ of $S$ with respect to $T$ in $\mathcal{G}$.
3: Initialize $\mathbf{C} = \varnothing$, a waiting queue $\mathcal{Q} = [\,]$, and a set $\mathcal{H} = \varnothing$,
4: **for** $\alpha \in sib(S) \cup ch(S)$ **do**
5:     add $(\alpha, S, \alpha)$ to the end of $\mathcal{Q}$,
6: **while** $\mathcal{Q} \neq \varnothing$ **do**
7:     take the first element $(\alpha, \phi, \tau)$ out of $\mathcal{Q}$ and add it to $\mathcal{H}$;
8:     **if** $\tau = T$ **then**
9:         add $\alpha$ to $\mathbf{C}$, and remove from $\mathcal{S}$ all triples where the first element is $\alpha$;
10:     **else**
11:         **for** each node $\beta$ in $\mathcal{G}$ **do**
12:             **if** $\tau \rightarrow \beta$ or $\tau - \beta$ **then**
13:                 **if** $\tau \rightarrow \beta$ or $\phi$ is not adjacent with $\beta$ or $\tau$ is the endnode **then**
14:                     **if** $\beta$ and $S$ are not adjacent **then**
15:                         **if** $(\alpha, \tau, \beta) \notin \mathcal{H}$ and $(\alpha, \tau, \beta) \notin \mathcal{Q}$ **then**
16:                             add $(\alpha, \tau, \beta)$ to the end of $\mathcal{Q}$,
17: **return** $\mathbf{C}$

---

## D.2 ADDITIONAL IDENTIFICATION RESULTS

In Zuo et al. (2022), the authors discuss that in the root node case (when the sensitive attribute do not have any parent in an MPDAG), any ancestral relationship between the sensitive attribute and other attribute is definite in an MPDAG. Here, we claim that the root node assumption is sufficient but not necessary and provide a complete graphical criterion under which any ancestral relationship with regards to a specific vertex is definite on an MPDAG in Theorem D.2.

We first introduce a technical lemma in Lemma D.1 that is useful in the proof of Theorem D.2

**Lemma D.1.** *Let $X$ and $W$ be two distinct vertices in an MPDAG $\mathcal{G} = (\mathbf{V}, \mathbf{E})$. The ancestral relationship between $X$ and $W$ is definite if there is no proper chordless possibly causal path from $X$ to $W$ in $\mathcal{G}$ that starts with an undirected edge.*

*Proof.* Suppose for a contradiction that $W$ is a possible descendant of $A$, then there is a possibly causal path $p$ from $A$ to $W$. By Lemma A.2, a subsequence $p^{*}$ of $p$ forms a chordless possibly causal path from $A$ to $W$. Suppose $p^{*} = \langle A = V_0, ..., V_k = W \rangle$. As there is no proper chordless possibly causal path from $A$ to $W$ starts with undirected edge, node $A$ must be directed towards $V_1$ as $A \rightarrow V_1$. By Lemma A.1, $p^{*}$ is a causal path from $A$ to $W$ in $\mathcal{G}$. Therefore, $W$ is a definite descendant of $A$. The ancestral relationship between $X$ and $W$ is definite. $\qquad\square$

**Theorem D.2.** *Let $A$ be a vertex in an MPDAG $\mathcal{G} = (\mathbf{V}, \mathbf{E})$. The ancestral relationship between $A$ and any other vertex is definite if and only if there are no undirected edges connected to $A$ in $\mathcal{G}$.*

*Moreover, an arbitrary attribute $W$ is a definite descendant of $A$ if and only if there is a causal path from $A$ to $W$ in $\mathcal{G}$.*

*Proof.* First, we prove the sufficiency. If there is no undirected edges connected to $A$ in an MPDAG $\mathcal{G}$, for any vertex $W$, there is no proper chordless possibly causal path from $A$ to $W$ that starts with an undirected edge. According to Lemma D.1, the ancestral relationship between $A$ and any other vertex is definite. Next, we prove the necessity. If there is an undirected edge connected to $A$, such as $A - W$, in $\mathcal{G}$, then there exists a DAG $\mathcal{D}1 \in [\mathcal{G}]$ with $A \leftarrow W$ so that $W$ is the non-descendant of $A$ in $\mathcal{D}1$. Meanwhile, there exists a DAG $\mathcal{D}2 \in [\mathcal{G}]$ with $A \rightarrow W$ so that $W$ is the descendant of $A$ in $\mathcal{D}2$. In this way, the ancestral relationship between $A$ and the vertex $W$ is not definite. $\square$

## E  MMD IN THE FAIRNESS CONTEXT

The MMD (Maximum Mean Discrepancy) is used to measure the distance between two data distributions. It is accomplished by mapping each sample to a Reproducing Kernel Hilbert Space (RKHS) using the kernel embedding trick firstly and then comparing the samples using a Gaussian kernel. In our context, let $\hat{\mathbf{y}}_{\mathbf{a}} = (\hat{y}_a^1, ..., \hat{y}_a^{N_a})$ and $\hat{\mathbf{y}}_{\mathbf{a}'} = (\hat{y}_{a'}^1, ..., \hat{y}_{a'}^{N_{a'}})$ be the prediction of the samples under intervention $do(\mathbf{A} = \mathbf{a}, \mathbf{X}_{ad} = \mathbf{x}_{ad})$ and $do(\mathbf{A} = \mathbf{a}', \mathbf{X}_{ad} = \mathbf{x}_{ad})$, respectively, where $N_a$ and $N_{a'}$ are number of the generated interventional samples under two interventions respectively. Then we can express the $\text{MMD}^2$ between predictions across different interventions as

$$\text{MMD}^2(\hat{\mathbf{y}}_{\mathbf{a}}, \hat{\mathbf{y}}_{\mathbf{a}'}) = \left\| \frac{1}{N_a} \sum_{i=1}^{N_a} \varphi(\hat{y}_a^i) - \frac{1}{N_{a'}} \sum_{j=1}^{N_{a'}} \varphi(\hat{y}_{a'}^j) \right\|^2,$$

where $\varphi(\cdot)$ represents the mapping function to RKHS. After substituting the kernel functions for the inner products, then the squared MMD is:

$$\frac{1}{(N_a)^2} \sum_{i=1}^{N_a} \sum_{j=1}^{N_a} k(\hat{y}_a^i, \hat{y}_a^j) + \frac{1}{(N_{a'})^2} \sum_{i=1}^{N_{a'}} \sum_{j=1}^{N_{a'}} k(\hat{y}_{a'}^i, \hat{y}_{a'}^i) - \frac{2}{N_a N_{a'}} \sum_{i=1}^{N_a} \sum_{j=1}^{N_{a'}} k(\hat{y}_a^i, \hat{y}_{a'}^j),$$

where $k(\cdot)$ is a kernel function. The widely used one is the Gaussian RBF kernel $k(x, x') = exp(-\frac{1}{\sigma} \|x, x'\|^2)$, with bandwidth $\sigma$.

## F  APPLICABILITY TO OTHER FAIRNESS NOTIONS

Our proposed method is directly applicable to other interventional-based fairness notions, such as ones defined on total effect (TE) and controlled direct effect (CDE) (Pearl, 2009) and no proxy discrimination (Kilbertus et al., 2017). In this section, we provide a detailed review of the related fairness notions and examine the applicability or inapplicability of the proposed identification criteria to these notions. Similar to the main text, let $\mathbf{A}$, $Y$ and $\mathbf{X}$ represent the sensitive attributes, outcome of interest and other observable attributes, respectively. The prediction of $Y$ is denoted by $\hat{Y}$. Besides, as in Section 4.2, we denote the augmented MPDAG $\mathcal{G} = (\mathbf{V}, \mathbf{E})$ with $\hat{Y}$, where $\mathbf{V} = \mathbf{X} \cup \mathbf{A}$, by $\mathcal{G}^*$. TE of $\mathbf{A}$ on $\hat{Y}$ corresponds to when $\mathbf{X}_{ad} = \varnothing$. It is identifiable if and only if no pair of nodes $A \in \mathbf{A}$ and $V \in \mathbf{V} \backslash \mathbf{A}$ such that $A - V$ is in $\mathcal{G}$. CDE of $\mathbf{A}$ on $\hat{Y}$ corresponds to $\mathbf{X}_{ad} = \mathbf{X}$. It is always identifiable under our modeling strategy [6]. The causal query in no proxy discrimination, $P(\hat{Y}|do(\mathbf{P} = \mathbf{p}))$, where $\mathbf{P}$ is the proxy, corresponds to $\mathbf{A} = \mathbf{P}$ and $\mathbf{X}_{ad} = \varnothing$.

More specifically, we start with the total causal effect (TE), which is defined as follows:

**Definition F.1** (Total causal effect)**.** *(Pearl, 2009) The total causal effect of the value change of $\mathbf{A}$ from $\mathbf{a}$ to $\mathbf{a}'$ on $\hat{Y} = \hat{y}$ is given by*

$$TE(\mathbf{a}, \mathbf{a}') = P(\hat{Y} = \hat{y}|do(\mathbf{A} = \mathbf{a})) - P(\hat{Y} = \hat{y}|do(\mathbf{A} = \mathbf{a}')).$$

---

[6]This meets CDE identification criterion on an MPDAG proposed by Flanagan (2020, Theorem 5.4).

The fairness criterion defined on total causal effect claims that the prediction $\hat{Y}$ is fair with respect to the sensitive attribute $\mathbf{A}$ if $P(\hat{Y} = \hat{y}|do(\mathbf{A} = \mathbf{a})) = P(\hat{Y} = \hat{y}|do(\mathbf{A} = \mathbf{a}'))$ holds for all possible values of $\hat{y}$ and any value that $\mathbf{A}$ can take. This notion corresponds to the notion of interventional fairness where $\mathbf{X}_{ad} = \varnothing$. Under the proposed modelling technique on $\hat{Y}$, Proposition 4.1 implies that the term $P(\hat{Y} = \hat{y}|do(\mathbf{A} = \mathbf{a}))$ is identifiable if and only if no pair of nodes $A \in \mathbf{A}$ and $V \in \mathbf{V} \backslash \mathbf{A}$ such that $A - V$ is in $\mathcal{G}$. In this case, total causal effect can be expressed as $f(\hat{y}|do(\mathbf{a})) = \int f(\hat{y}|\mathbf{v}) \prod_{\mathbf{V_i} \subseteq \mathbf{V}} f(\mathbf{v_i}|pa(\mathbf{v_i}, \mathcal{G}))d\mathbf{v}'$, where $\mathbf{V}' = \mathbf{V} \backslash \mathbf{A}$, $(\mathbf{V_1}, ..., \mathbf{V_m}) = \text{PCO}(\mathbf{V}', \mathcal{G})$, with $pa(\mathbf{v_i}, \mathcal{G})$ representing values of $Pa(\mathbf{v_i}, \mathcal{G})$ that agree with $\mathbf{a}$.

**Definition F.2** (Controlled direct causal effect). *(Pearl, 2009) The controlled direct effect of the value change of $\mathbf{A}$ from $\mathbf{a}$ to $\mathbf{a}'$ on $\hat{Y} = \hat{y}$, while fixing all the other variables $\mathbf{X}$ to $\mathbf{x}$ is given by*

$$CDE(\mathbf{a}, \mathbf{a}', \mathbf{x}) = P(\hat{Y} = \hat{y}|do(\mathbf{A} = \mathbf{a}, \mathbf{X} = \mathbf{x})) - P(\hat{Y} = \hat{y}|do(\mathbf{A} = \mathbf{a}', \mathbf{X} = \mathbf{x})).$$

The fairness criterion defined on controlled direct causal effect claims that the prediction $\hat{Y}$ is fair with respect to the sensitive attribute $\mathbf{A}$ if $P(\hat{Y} = \hat{y}|do(\mathbf{A} = \mathbf{a}, \mathbf{X} = \mathbf{x})) = P(\hat{Y} = \hat{y}|do(\mathbf{A} = \mathbf{a}', \mathbf{X} = \mathbf{x}))$ holds for all possible values of $\hat{y}$ and any value that $\mathbf{A}$ and $\mathbf{X}$ can take. This notion corresponds to the notion of interventional fairness where $\mathbf{X}_{ad} = \mathbf{X}$. Proposition 4.1 implies that it is always identifiable under our modeling strategy and is given by $f(\hat{y}|do(\mathbf{a}), do(\mathbf{x})) = f(\hat{y}|\mathbf{a}, \mathbf{x})$.

**Definition F.3** (No proxy discrimination). *(Kilbertus et al., 2017) A predictor $\hat{Y}$ exhibits no proxy discrimination based on the proxy $\mathbf{P}$ if for any value that $\mathbf{P}$ can take,*

$$P(\hat{Y} = \hat{y}|do(\mathbf{P} = \mathbf{p})) = P(\hat{Y} = \hat{y}|do(\mathbf{P} = \mathbf{p}')).$$

This fairness criterion corresponds to the interventional fairness where $\mathbf{A} = \mathbf{P}$ and $\mathbf{X}_{ad} = \varnothing$. Under the proposed modelling technique on $\hat{Y}$, Proposition 4.1 implies that the term $P(\hat{Y} = \hat{y}|do(\mathbf{P} = \mathbf{p}))$ is identifiable if and only if there exists no pair of nodes $P \in \mathbf{P}$ and $V \in \mathbf{V} \backslash \mathbf{P}$ such that $P - V$ is in $\mathcal{G}$. In this case, the causal query $f(\hat{y}|do(\mathbf{p}))$ can be expressed as $f(\hat{y}|do(\mathbf{p})) = \int f(\hat{y}|\mathbf{v}) \prod_{\mathbf{V_i} \subseteq \mathbf{V}} f(\mathbf{v_i}|pa(\mathbf{v_i}, \mathcal{G}))d\mathbf{v}'$, where $\mathbf{V}' = \mathbf{V} \backslash \mathbf{P}$, $(\mathbf{V_1}, ..., \mathbf{V_m}) = \text{PCO}(\mathbf{V}', \mathcal{G})$, with $pa(\mathbf{v_i}, \mathcal{G})$ representing values of $Pa(\mathbf{v_i}, \mathcal{G})$ that agree with $\mathbf{p}$.

**Definition F.4** (Path-specific causal effect). *(Avin et al., 2005) Given a causal path set $\pi$, the $\pi$-specific effect of the value change of $\mathbf{A}$ from $\mathbf{a}$ to $\mathbf{a}'$ on $\hat{Y} = \hat{y}$ through $\pi$ is given by*

$$PE_\pi(\mathbf{a}, \mathbf{a}') = P(\hat{Y} = \hat{y}|do(\mathbf{A} = \mathbf{a}'|\pi), do(\mathbf{A} = \mathbf{a}|\bar{\pi})) - P(\hat{Y} = \hat{y}|do(\mathbf{A} = \mathbf{a})),$$

*where $P(\hat{Y} = \hat{y}|do(\mathbf{A} = \mathbf{a}'|\pi), do(\mathbf{A} = \mathbf{a}|\bar{\pi}))$ represents the post-intervention distribution of $\hat{Y}$ where the effect of intervention $do(\mathbf{A} = \mathbf{a}')$ is transmitted only along $\pi$ while the effect of intervention $do(\mathbf{A} = \mathbf{a})$ is transmitted along the other paths.*

The fairness criterion defined on path-specific causal effect claims that a predictor $\hat{Y}$ achieves path-specific causal fairness with respect to the sensitive attribute $\mathbf{A}$ and the path set $\pi$ if $PE_\pi(\mathbf{a}, \mathbf{a}') = 0$ holds for all possible value of $\hat{y}$ and any value that $\mathbf{A}$ can take. However, the term in $PE_\pi(\mathbf{a}, \mathbf{a}')$, $P(\hat{Y} = \hat{y}|do(\mathbf{A} = \mathbf{a}'|\pi), do(\mathbf{A} = \mathbf{a}|\bar{\pi}))$, also known as a nested counterfactual query, cannot be identified by the formula in Proposition 4.1 on MPDAG. Similarly, our proposed method cannot deal with the counterfactual fairness or path-specific counterfactual fairness defined below. That is because the ongoing challenge of the identification of the counterfactual queries in these notions on MPDAGs, specifically, $P(\hat{Y}_{\mathbf{A} \leftarrow \mathbf{a}'}(U) = y|\mathbf{X} = \mathbf{x}, \mathbf{A} = \mathbf{a})$ and $P(\hat{Y}_{\mathbf{A}=\mathbf{a}'|\pi, \mathbf{A}=\mathbf{a}|\bar{\pi}}|\mathbf{O} = \mathbf{o})$.

**Definition F.5** (Counterfactual fairness). *(Kusner et al., 2017, Definition 5) We say the prediction $\hat{Y}$ is counterfactually fair if under any context $\mathbf{X} = \mathbf{x}$ and $\mathbf{A} = \mathbf{a}$,*

$$P(\hat{Y}_{\mathbf{A} \leftarrow \mathbf{a}}(U) = \hat{y}|\mathbf{X} = \mathbf{x}, \mathbf{A} = \mathbf{a}) = P(\hat{Y}_{\mathbf{A} \leftarrow \mathbf{a}'}(U) = \hat{y}|\mathbf{X} = \mathbf{x}, \mathbf{A} = \mathbf{a}),$$

*for all possible value of $\hat{y}$ and any value attainable by $\mathbf{A}$.*

**Definition F.6** (Path-specific counterfactual fairness). *(Chiappa, 2019; Wu et al., 2019b) Given a factual condition $\mathbf{O} = \mathbf{o}$ where $\mathbf{O} \subseteq \{\mathbf{A}, \mathbf{X}, \hat{Y}\}$ and a causal path set $\pi$, predictor $\hat{Y}$ achieves the path-specific counterfactual fairness if*

$$P(\hat{Y}_{\mathbf{A}=\mathbf{a}'|\pi, \mathbf{A}=\mathbf{a}|\bar{\pi}} = \hat{y}|\mathbf{O} = \mathbf{o}) = P(\hat{Y}_{\mathbf{A}=\mathbf{a}} = \hat{y}|\mathbf{O} = \mathbf{o}),$$

*for all possible value of $\hat{y}$ and any value attainable by $\mathbf{A}$.*

# G    SUPPLEMENTARY EXPERIMENTAL RESULTS

## G.1    CAUSAL GRAPHS FOR ONE SIMULATION

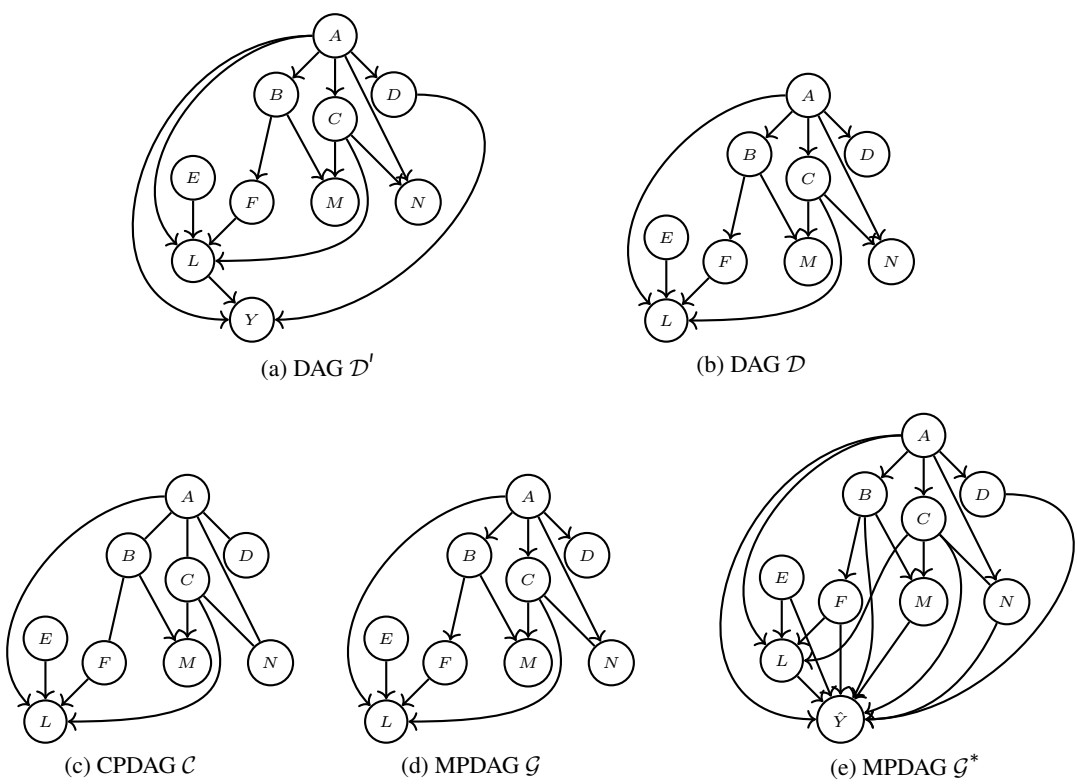

Figure 10: (a) is one of the generated DAG $\mathcal{D}^I$ with 10 nodes and 15 directed edges. The randomly selected sensitive attribute is represented by $A$ and the outcome attribute is $Y$; (b) is the subgraph of $\mathcal{D}^I$ over all of the observable variables except $Y$, denoted by DAG $\mathcal{D}$; (c) is the corresponding CPDAG $\mathcal{C}$ such that $\mathcal{D} \in [\mathcal{C}]$; (d) With the background knowledge $\{A \rightarrow B, A \rightarrow C, A \rightarrow D, A \rightarrow N\}$, we can obtain the MPDAG $\mathcal{G}$, such that $\mathcal{D} \in [\mathcal{G}]$. The definite non-descendant of $A$ can be found to be $\{E\}$ by leveraging Algorithm 3; (e) is the extended-$\mathcal{G}$ with $\hat{Y}$, denoted as $\mathcal{G}^*$.

## G.2 Causal Graphs for the UCI Student Dataset

The causal graphs for the UCI Student Dataset is provided in Figure 11. The attribute information can be found at `https://archive.ics.uci.edu/ml/datasets/Student+Performance`.

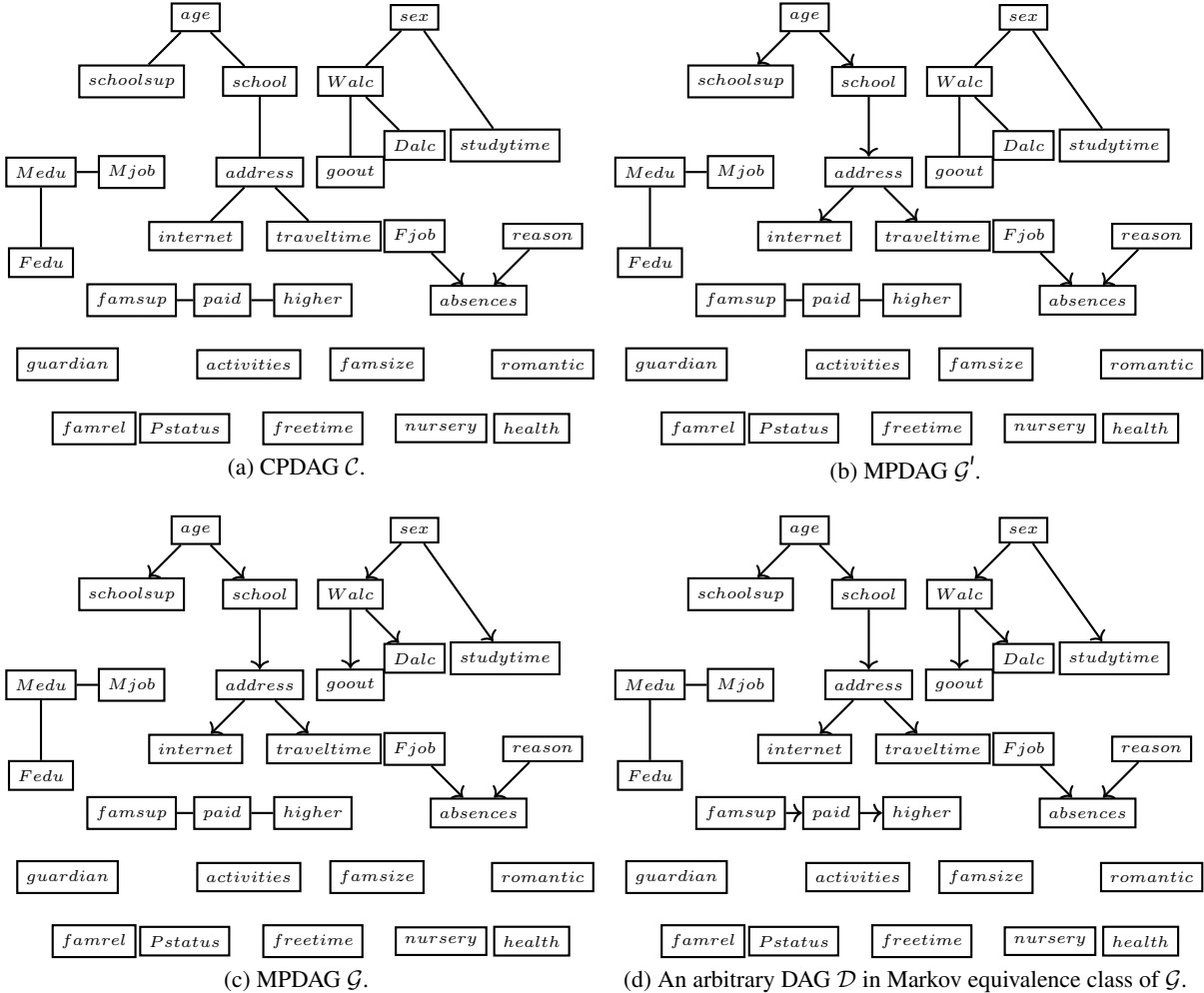

Figure 11: The causal graphs for Student dataset. (a) is the CPDAG $\mathcal{C}$ learnt from the observational data; (b) Given the background knowledge that the age is the parent of schoolsup and school, we can have the MPDAG $\mathcal{G}^I$ by leveraging Meek's rule; (c) Considering the additional background knowledge constraint that the sensitive attribute that cannot have any parent in the graph, we can obtain the MPDAG $\mathcal{G}$; (d) is an arbitary DAG $\mathcal{D}$ in the Markov equivalence class of $\mathcal{G}$.

## G.3 More Experimental Details on the UCI Student Dataset

We employ the GES structure learning algorithm (Chickering, 2002a) implemented in TETRAD (Ramsey et al., 2018), a general causal discovery software, to learn the CPDAG from the dataset, excluding the target attribute Grade. After uploading the preprocessed data, we can learn the CPDAG $\mathcal{C}$. The evolution of the CPDAG to MPDAG is shown in Figure 11 in Appendix G.2. Our experiments are carried out on the MPDAG $\mathcal{G}$ in Figure 11c. In this dataset, the definite descendants of sex can be identified as {Walc, goout, Dalc, studytime} and all the other nodes are definite non-descendants of sex.

The dataset is divided into three sets: training, validation, and test, in an 8:1:1 ratio. Interventional data on sex = female and sex = male in the test set is obtained via splitting the test data by different values of sex. On the other hand, the interventional data for training and validation sets are generated according to the causal identification formula. To generate interventional data, we first apply Algorithm 2 to the MPDAG $\mathcal{G}$ in Figure 11c. This gives us an ordered list of the bucket decomposition of vertices in $\mathcal{G}$: {{age}, {schoolsup}, {school}, {address}, {internet}, {traveltime}, {sex}, {Walc}, {studytime}, {Dalc}, {goout}, {Mjob, Medu, Fedu}, {Fjob}, {reason}, {famsup, paid, higher}, {guardian}, {activities}, {famsize}, {romantic}, {famrel}, {Pstatus}, {freetime}, {nursery}, {health}, {absences}}. For each pair of buckets and its parents, we fit a conditional multivariate distribution using mixture density networks Bishop (1994) for continuous variables and probability contingency tables for discrete variables. For example, Figure 12 demonstrates the fitting performance on the attributes school, address, internet and traveltime. Based on the fitted conditional distribution, we generate 395 interventional data points for sex = female and sex = male, respectively. The proportion for training and validation interventional data is 8:2. Additionally, in our $\epsilon$-IFair model, the $\lambda$ is set to be $[0, 1, 40, 100, 130, 175, 250]$. For each model, we run it 20 times with different seeds and report the average results in the main section.

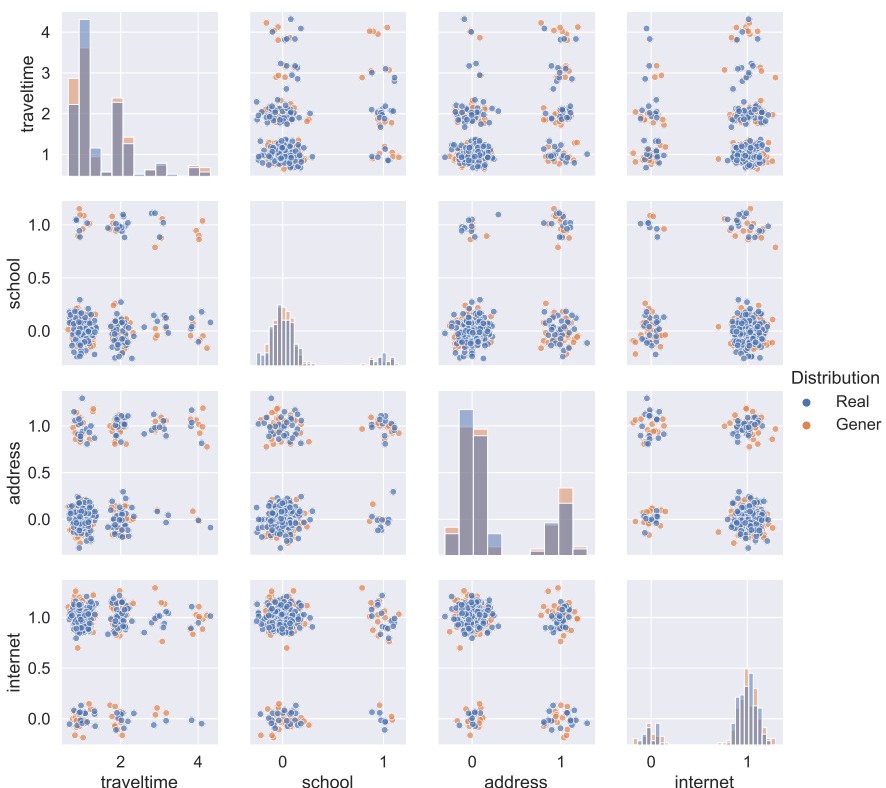

Figure 12: The fitting performance on the attributes school, address, internet and traveltime in the Student dataset. 'Real' represents the observational data where sex = male, while 'Gener' represents the generated interventional data on sex = male with the fitted conditional density. To better show the fitting performance, a very small amount of noise is added to discrete variables.

## G.4 CAUSAL GRAPHS FOR THE CREDIT RISK DATASET

The causal graphs for the Credit Risk Dataset is provided in Figure 13. The attribute information can be found at `https://www.kaggle.com/datasets/laotse/credit-risk-dataset`.

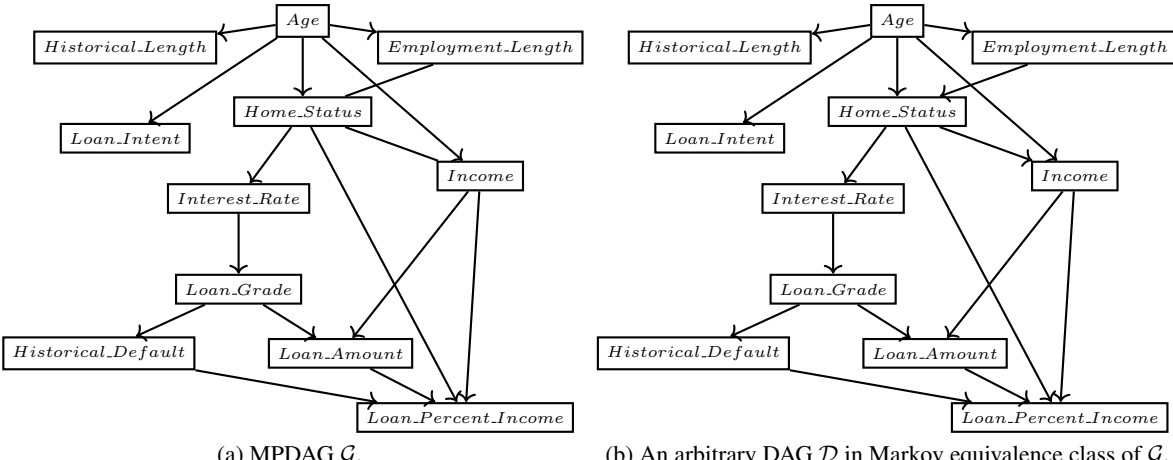

(a) MPDAG $\mathcal{G}$.   (b) An arbitrary DAG $\mathcal{D}$ in Markov equivalence class of $\mathcal{G}$.

Figure 13: The causal graphs for the Credit Risk dataset. (a) is the learned MPDAG $\mathcal{G}$ from the observational data with tier ordering constraint: Age is placed in the first tier, while all other variables are in the second tier; (b) is an arbitrary DAG $\mathcal{D}$ in the Markov equivalence class of $\mathcal{G}$.

## G.5 MORE EXPERIMENTAL DETAILS ON THE CREDIT RISK DATASET

To obtain an MPDAG over all variables except Loan_status, we utilize the GES structure learning algorithm (Chickering, 2002a) implemented in TETRAD. When learning the graph, we consider the tier ordering as background knowledge, where Age is placed in the first tier while all other variables are in the second tier. The resulting MPDAG $\mathcal{G}$ is depicted in Figure 13a in Appendix G.4. Since there are no definite non-descendants of Age in this dataset, an `IFair` model is not applicable. The process of interventional data generation and model training follows a similar approach as Section 5.2.1.

After preprocessing, we focus on two specific age groups: 23 and 30. The age group of 23 consists of 3,413 records, while the age group of 30 has 1,126 records. We divide the whole dataset of 4539 records into three sets: training, validation, and test, using an 8:1:1 ratio. Interventional data for the test set is obtained by splitting the test data by Age = 23 and Age = 30. Similarly, the interventional data for training and validation sets are generated according to the causal identification formula. To generate interventional data, we first apply Algorithm 2 to the MPDAG $\mathcal{G}$ in Figure 13a. This provides us with an ordered list of the bucket decomposition of vertices in $\mathcal{G}$: {{Age}, {Historical_Length}, {Loan_Intent}, {Income, Home_Status, Employment_Length}, {Interest_Rate}, {Loan_Grade}, {Historical_Default}, {Loan_Amount}, {Loan_Percent_Income}. For each pair of buckets and its parents, we fit a conditional multivariate distribution using mixture density network Bishop (1994) for continuous variables and probability contingency tables for discrete variables. As an example, Figure 14 demonstrates the fitting performance on the attribute Loan_Percent_Income and its parents Income, Loan_Amount, Home_Status, Historical_Default. Based on the fitted conditional distribution, we generate 4539 interventional data points for age = 23 and age = 30, respectively. The proportion for training and validation interventional data is 8:2. Additionally, in our $\epsilon$-`IFair` model, the $\lambda$ is set to be $[2, 4, 6, 10, 15, 150]$. For each model, we run it 10 times with different seeds and report the average results in the main section.

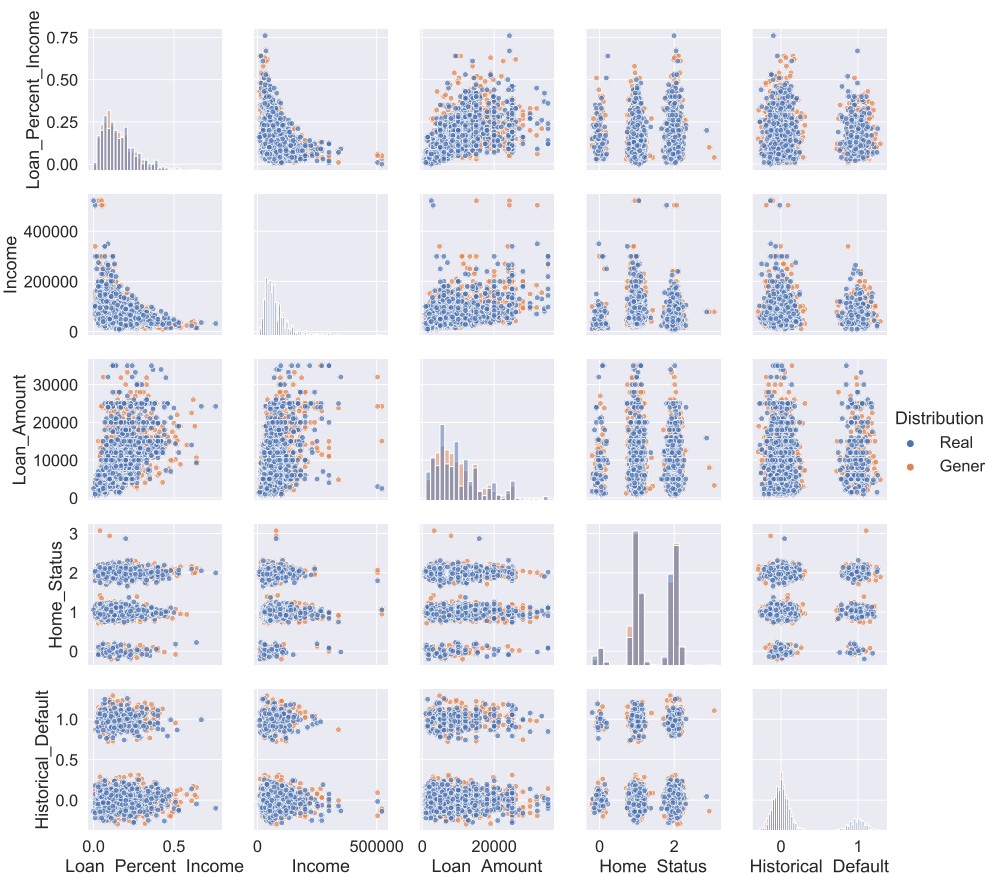

Figure 14: The fitting performance on the attribute Loan_Percent_Income and its parents Income, Loan_Amount, Home_Status, Historical_Default in the Credit Risk dataset. 'Real' represents the observational data where Age = 30, while 'Gener' represents the generated interventional data on Age = 30 by the fitted conditional density. To better show the fitting performance, a very small amount of noise is added to discrete variables.

## G.6 EXPERIMENTS INVOLVING NON-EMPTY ADMISSIBLE VARIABLE SET

For graphs with node $d = \{5, 10, 20, 30\}$ and the corresponding edge $s = \{8, 20, 40, 60\}$, with the same basic settings and evaluation metrics as Section 5.1, we randomly choose one, one, two and three admissible variables for each graph setting respectively. The admissible variable is intervened to be the mean of that variable in the observational data. The accuracy fairness trade-off plot is shown in Figure 15. We can see that it follows a similar trend as Section 5.1. For certain values of $\lambda$, we can simultaneously achieve low RMSE comparable to `Full` model and low unfairness comparable to `IFair`.

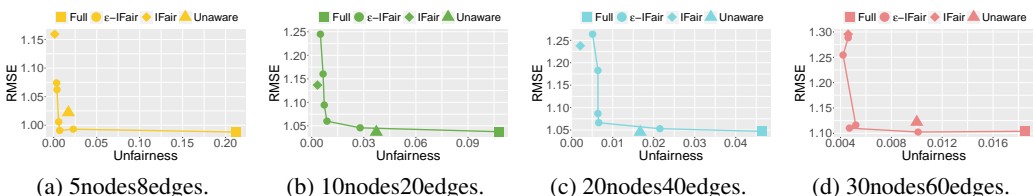

| (a) 5nodes8edges. | (b) 10nodes20edges. | (c) 20nodes40edges. | (d) 30nodes60edges. |

Figure 15: Accuracy fairness trade-off for the case involving non-empty admissible variable set.

## G.7 EXPERIMENTS BASED ON MORE COMPLICATED STRUCTURAL EQUATIONS

To demonstrate the generality of our method, we generate each variable $X_i$ based on a non-linear structural equation given by:

$$X_i = f_i(pa(X_i) + \epsilon_i), i = 1, ..., n, \tag{5}$$

where the causal mechanism $f_i$ is randomly selected from functions such as *linear*, *sin*, *cos*, *tanh*, *sigmoid*, and their combinations. The noise term $\epsilon_i$ is sampled from *Gaussian* distribution. With the same basic settings and evaluation metrics as described in Section 5.1, we present the accuracy fairness trade-off plot in Figure 16 in solid line. We can see that it follows a similar trend as Section 5.1. For some $\lambda$, we can simultaneously achieve low RMSE comparable to `Full` model and low unfairness comparable to `IFair`. Additionally, we also include the results obtained when the interventional data is generated from the ground-truth structural equations in dotted line. The discrepancy between the two lines reflects the bias introduced by the approximation of conditional densities. The minimal discrepancy observed in Figure 16 suggests that the conditional densities are well fitted. Specifically, we analyze model robustness on the approximation of conditional densities on the linear data in Appendix G.10.

## G.8 EXPERIMENTAL ANALYSIS ON MODEL PERFORMANCE WITH VARYING AMOUNT OF GIVEN DOMAIN KNOWLEDGE

The performance of baseline models `Full` and `Unaware` remain unaffected by the amount of background knowledge. When all ancestral relations between the sensitive attribute and other variables are definite, the performance of `IFair` model remains consistent regardless of the additional background knowledge. On the other hand, for the $\epsilon$-`IFair` model, once the causal effect on an MPDAG is identifiable, additional background knowledge does not theoretically impact the performance at a given $\lambda$ since the same causal identification formula can be applied. However, in practical experiments, slight variations may arise due to the error in fitting of different conditional densities. To vary the amount of given background knowledge, we adjust the proportion of the undirected edges' true orientation in a CPDAG that is considered as the background knowledge. In the '10nodes20edges' setting, we increase the proportion from 0.1, 0.3, 0.6 to 1.0. The 'BK (%)' value quantifies the proportion of the undirected edges' true orientation that is considered background knowledge. The trade-off plots in Figure 17 illustrate this relationship between the amount of background knowledge and the resulting tradeoff between fairness and accuracy.

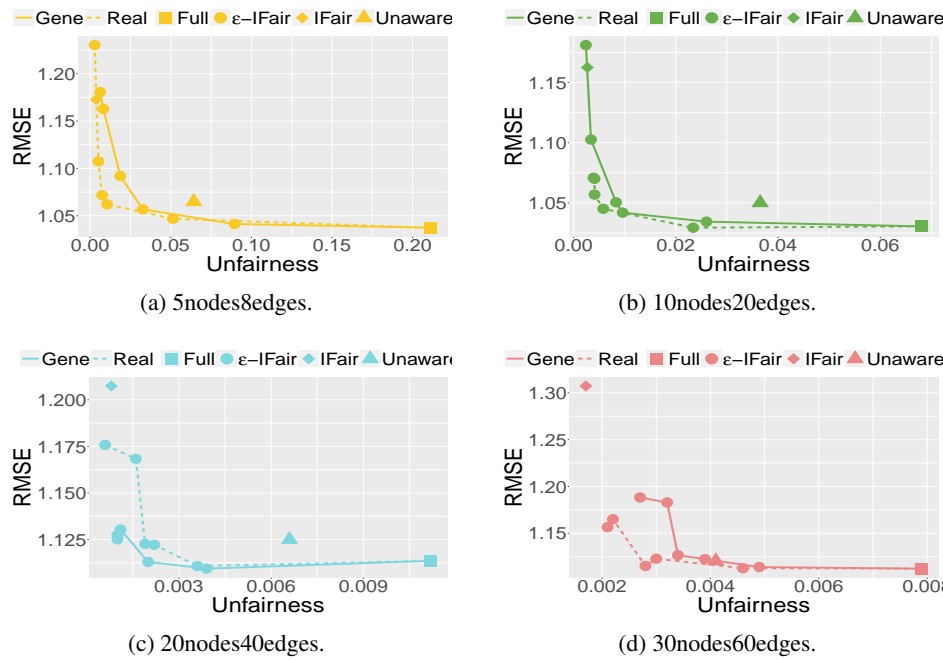

Figure 16: Accuracy fairness trade-off plots for the non-linear synthetic data. The solid line ('Gene') depicts the results when the interventional data is generated by the fitted conditional densities, while the dotted line ('Real') depicts the results when the interventional data is generated by the ground-truth structural equations.

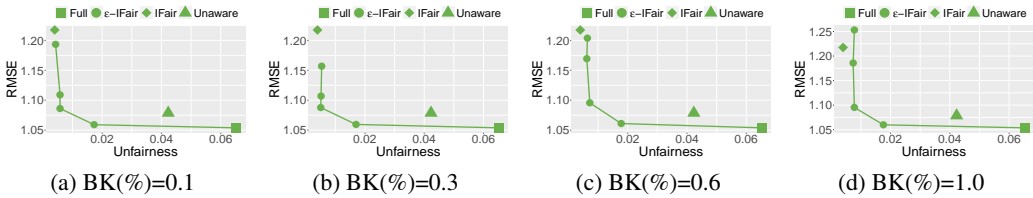

Figure 17: Accuracy fairness trade-off plots with varying amount of given background knowledge on the graph setting '10nodes20edges'.

### G.9 EXPERIMENTAL ANALYSIS ON MODEL ROBUSTNESS ON CAUSAL DISCOVERY ALGORITHMS

In Section 5.1, we derive the ture CPDAG directly from the known DAG without utilizing any causal discovery algorithm. However, in practical scenarios where the true DAG is unknown, the CPDAG can only be obtained from causal discovery algorithms. To assess the robustness of our model with respect to causal discovery algorithms, we employed the Greedy Equivalence Search (GES) procedure Chickering (2002a) to learn the corresponding CPDAG from the synthetic data. Under the '10nodes20edges' graph setting, Figure 18a provides a comparison between the results obtained when the CPDAG is derived from the true DAG and when it is learned from the observational data using the GES algorithm. Both sets of results exhibit a similar trend, indicating that our model is robust to the GES algorithm for causal discovery. Additionally, for reference, we show each case with the scenario where the interventional data is generated from the ground-truth structural equation in Figure 18b and Figure 18c respectively. Similarly, we obtain analogous results for the '20nodes40edges' graph setting, which are displayed in Figure 18d, Figure 18e and Figure 18f.

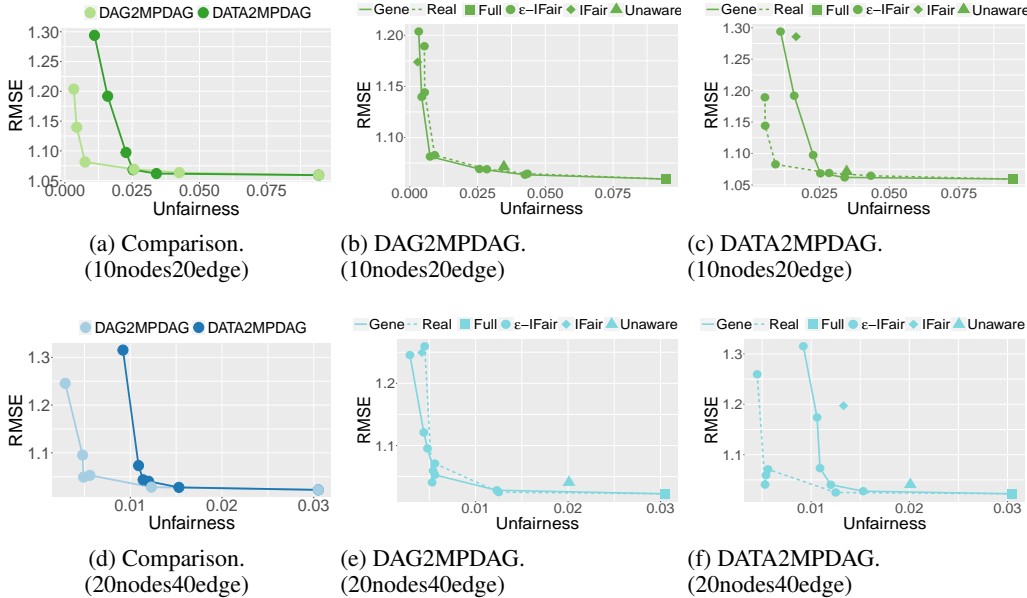

Figure 18: Accuracy fairness trade-off plots analyzing model robustness on causal discovery algorithms on the graph settings '10nodes20edges' and '20nodes40edges'. In (b),(c),(e) and (f), the solid line ('Gene') depicts the results when the interventional data is generated by the fitted conditional densities, while the dotted line ('Real') depicts the results when the interventional data is generated by the ground-truth structural equations.

## G.10 EXPERIMENTAL ANALYSIS ON MODEL ROBUSTNESS ON CONDITIONAL DENSITIES APPROXIMATION

Using the same experimental setting as described in Section 5.1, we present the accuracy fairness trade-off plot for linear synthetic data in Figure 19 in a solid line. In order to analyze the extent to which bias is introduced by approximating conditional densities, we also include the results obtained when the interventional data is generated from the ground-truth structural equations in Figure 19 in dotted line. The discrepancy between the two lines reflect the bias introduced by fitting the conditional densities. The minimal discrepancy observed in Figure 19 suggests that the conditional densities are well fitted.

## G.11 EXPERIMENTAL ANALYSIS ON MODEL PERFORMANCE ON UNIDENTIFIABLE CASES

Under the graph setting '10nodes20edges', utilizing the same data generation process as in Section 5.1, we conduct experiments under a scenario where no explicit background knowledge is added for fairness identification. In cases where the MPDAG is unidentifiable, we apply the methodology proposed in Section 4.2. We select four exemplar unidentifiable MPDAGs, encompassing 2, 4, 2 and 9 valid MPDAGs separately. Figure 20 shows the prediction and fairness results for each of them, which is denoted as 'Unidentifiable'. Additionally, we compare these results with ones trained on the ground-truth MPDAGs, wherein the causal effect from $A$ to $\hat{Y}$ is identifiable, which is denoted by 'Identifiable'. As depicted in Figure 20, we can see that the trade-off between accuracy and fairness persists in these unidentifiable cases. Interestingly, compared with the prediction learnt solely on the ground-truth identifiable MPDAG, the prediction learnt on all possible MPDAGs does not necessarily deteriorate when achieving equivalent levels of unfairness. This could be attributed to the fact that they are penalizing different measures on fairness.

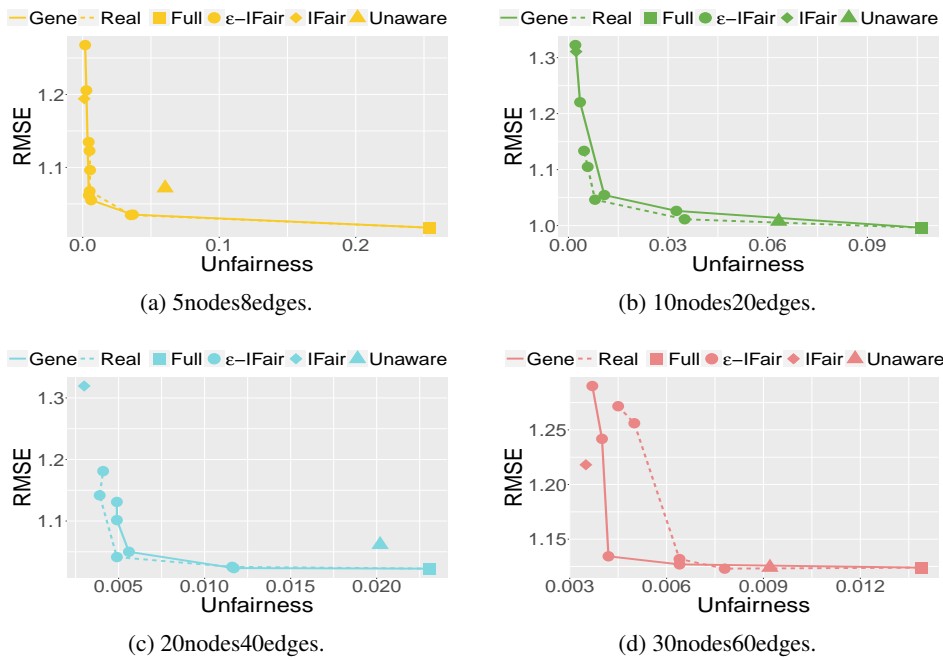

Figure 19: Accuracy fairness trade-off plots analyzing model robustness on the approximation of conditional densities for the linear synthetic data. The solid line ('Gene') depicts the results when the interventional data is generated by the fitted conditional densities, while the dotted line ('Real') depicts the results when the interventional data is generated by the ground-truth structural equations.

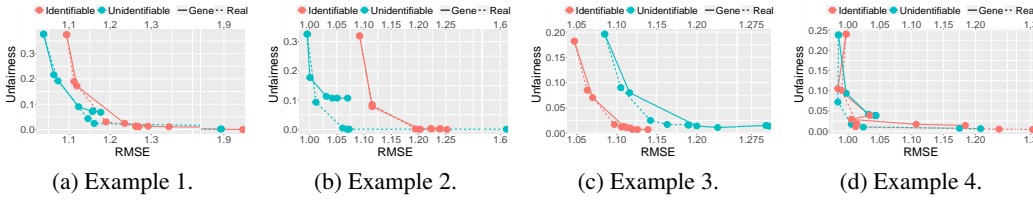

Figure 20: Accuracy fairness trade-off plots analyzing model performance on four exemplar unidentifiable cases, which is denoted as 'Unidentifiable'. 'Identifiable' denotes the results learnt solely from the ground-truth MPDAG with identifiable results. The solid line ('Gene') depicts the results when the interventional data is generated by the fitted conditional densities, while the dotted line ('Real') depicts the results when the interventional data is generated by the ground-truth structural equations.

# H    ADDITIONAL RELATED WORK ON CAUSAL EFFECT IDENTIFICATION ON MPDAGS

Identifying causal effects from a causal graph that represents observational data under the assumption of causal sufficiency is a fundamental problem. When the causal DAG is known, it is possible to identify and estimate all causal effects from the observational data (see e.g. Pearl (1995); Pearl & Robins (1995); Robins (1986); Galles & Pearl (1995). In this study, we focus on identifying causal effects in the context of an MPDAG $\mathcal{G}$. We consider a causal effect to be identifiable in an MPDAG $\mathcal{G}$ if the interventional density of the response can be uniquely computed from $\mathcal{G}$. The precise definition of identifiability of causal effects is provided in Definition H.1.

**Definition H.1** (Identifiability of Causal Effects). *(Perkovic, 2020, Definition 3.1) In an MPDAG* $\mathcal{G} = (\mathbf{V}, \mathbf{E})$*, let* $\mathbf{X}$ *and* $\mathbf{Y}$ *be disjoint node sets. The causal effect of* $\mathbf{X}$ *on* $\mathbf{Y}$ *is identifiable in* $\mathcal{G}$ *if* $f(\mathbf{y}|do(\mathbf{x}))$ *is uniquely computable from any observational density consistent with* $\mathcal{G}$*.*

*Hence, there exist no two DAGs* $\mathcal{D}1$ *and* $\mathcal{D}2$ *in* $[\mathcal{G}]$ *such that*

1. $f_1(\mathbf{v}) = f_2(\mathbf{v}) = f(\mathbf{v})$*, where* $f$ *is an observational density consistent with* $\mathcal{G}$*, and*

2. $f_1(\mathbf{y}|do(\mathbf{x})) \neq f_2(\mathbf{y}|do(\mathbf{x}))$*, where* $f_1(\cdot|do(\mathbf{x}))$ *and* $f_2(\cdot|do(\mathbf{x}))$ *are interventional den-sityies consistent with* $\mathcal{D}1$ *and* $\mathcal{D}2$ *respectively.*

In recent years, there has been extensive research focused on the identification of causal effects in MPDAGs. One notable contribution in this field is the generalized adjustment criterion proposed by Perković et al. (2015); Perkovi et al. (2017); Perkovi et al. (2018), which provides a sufficient condition for identifying causal effects. However, it is important to note that this criterion is not necessary for causal effect identification. To address this limitation, Perkovic (2020) introduces a graphical criterion that is both necessary and sufficient for the identification of causal effects in MPDAGs. In addition to the above developments, IDA algorithms and joint-IDA Maathuis et al. (2009); Nandy et al. (2017); Perkovic et al. (2017); Witte et al. (2020); Fang & He (2020); Liu et al. (2020) have been proposed for identifying the total effect of a variable $A$ on the response $Y$ in an MPDAG $\mathcal{G}$. These algorithms consider the orientation configurations of edges connected to variable $A$ and enumerate a set of MPDAGs where the total effect is identified. Specifically, Guo & Perkovic (2021) provide a characterization of the minimal additional edge orientations required to identify a given total effect.

## I  DISCUSSION ON ACCURACY-FAIRNESS TRADE-OFF

The trade-off between accuracy and fairness has been highlighted in numerous studies on algorithmic fairness (Martinez & Bertran, 2019; Zhao & Gordon, 2019; Menon & Williamson, 2018; Zliobaite, 2015; Chen et al., 2018; Wei & Niethammer, 2022). These studies highlight that improving fairness measures often comes at the cost of reduced accuracy in machine learning models. However, it is important to note that the existence of an accuracy-fairness trade-off is not universal and depends on the specific data setting. Several recent works have challenged the notion that accuracy must always decrease as fairness increases (Dutta et al., 2020; Friedler et al., 2021; Yeom & Tschantz, 2018). For instance, in the synthetic dataset discussed in Section 5.1, we observed an accuracy fairness trade-off, where improving fairness measures resulted in a decrease in accuracy. However, such a trade-off was not apparent in Figure 15d. The authors in (Wick et al., 2019; Sharma et al., 2020; Dutta et al., 2020) provide theoretical and empirical insights into when the trade-off exists and when it does not. These findings suggest that the trade-off relationship between accuracy and fairness is context-dependent. It highlights the need for further research to better understand the conditions under which the accuracy fairness trade-off arises and identify strategies to mitigate or overcome it.

