# OpenReview forum: "Interventional Fairness on Partially Known Causal Graphs: A Constrained Optimization Approach"
_ICLR.cc/2024/Conference — ICLR 2024 poster_

### Official Review · Reviewer_U24x · 2023-10-26

**Soundness:** 3 good
**Presentation:** 3 good
**Contribution:** 2 fair
**Rating:** 6
**Confidence:** 4

**Summary:**

The paper deals with the problem of learning an interventionally fair classifier when the underlying causal graph is not available. For this purpose, the authors propose to first apply a causal discovery algorithm, resulting in a maximally partially directed graph (MPDAG). Then, the authors add the predictor as an additional node to the causal graph and use identifiability theory for MPDAGs to estimate discrepancies with respect to interventional fairness, which is then added as a penalty to the predictor loss. The method is evaluated on synthetic and real-world data.

**Strengths:**

- The paper deals with an important topic, as the causal graph of the underlying data-generating process is often unknown in practice
- The method is theoretically sound, identifiability guarantees and proofs are provided
- The experimental results indicate the effectiveness of the proposed method

**Weaknesses:**

- The idea of adding a fairness penalty to the prediction objective and using a trade-off parameter is not novel but has been explored in previous papers with numerous fairness notions (e.g., Quinzan et al. 2022, Frauen et al. 2023). From my understanding, the novelty is combining these ideas with causal discovery and identifiability theory for MPDAGs.
- Related to the previous point, the paper seems to combine these results in a rather straightforward manner, with limited novel ideas. However, the corresponding theoretical results (e.g., Theorem 4.1) seem sound and no previous work seems to have applied MPDAG identifiability theory to interventional fairness (which, in my opinion, is enough novelty for recommending acceptance, but also justifies why I am hesitant to give a higher score).
- Applicability in practice: The proposed method seems to be difficult to employ in practice. First, one has to run a causal discovery algorithm to obtain an MPDAG, and then perform conditional density estimation to obtain the fairness penalty. Furthermore, there is no way of automatically choosing the trade-off parameter (even though this point is not specifically a drawback of the method in this paper).
-  The authors hint at the possibility of extending their approach to other fairness notions. I think the paper would benefit from a more detailed review of related fairness notions in the appendix (e.g., total/direct/indirect effects, path-specific effects, counterfactual fairness), and how the method could (or could not) be extended

--- Post rebuttal response ---
I thank the authors for their rebuttal. After reading the comments, I continue to be positive about the paper.

**Questions:**

- In the case of non-identification, the authors propose to replace the penalty term with a sum over all possible MPDAGs by directing the relevant edges. Can the authors provide some intuition/experiments on how this affects the prediction performance? I could imagine that the prediction performance could suffer if the objective is penalized with a large sum of MPDAGs.

---

> ### Author Response · Authors · 2023-11-20
> **Official Comment by Authors (1/2)**
>
> Thank you for your constructive comments. We greatly appreciate your thoughtful input and address your comments point by point as follows. We have also tried our best to revise the manuscript according to your suggestions and the revision is uploaded accordingly.
>
> > W1: The idea of adding a fairness penalty to the prediction objective and using a trade-off parameter is not novel but has been explored in previous papers with numerous fairness notions (e.g., Quinzan et al. 2022, Frauen et al. 2023). From my understanding, the novelty is combining these ideas with causal discovery and identifiability theory for MPDAGs.
>
> Thanks for raising this concern. The novelty of our approach mainly lies in that we provide an effective and practical method to achieve intervention fairness, when the underlying causal DAG is unknown. The term 'effective and practical' here refers to our method's ability to maximize the prediction accuracy and manage the trade-off between accuracy and fairness comparing with the existing method. In order to achieve this, we employ a causal discovery algorithm to ascertain a MPDAG in cases where the underlying DAG is unknown. Then we propose a modelling technique on the prediction $\hat{Y}$, such that the resulting augmented graph remains an MPDAG (as detailed in Theorem 4.1). It is important to emphasize that this result is of independent interest as it establishes a general modeling method applicable to any problem and enables various causal inference tasks beyond merely identifying causal effects formally within such augmented MPDAGs. To ascertain the degree of interventional fairness on the augmented MPDAG, we delve into the results of causal effect identification on MPDAGs. This investigation necessitates the formulation of a constrained optimization problem, which achieves the desired balance between accuracy and fairness. Additionally, it is crucial to emphasize that our proposed method extends beyond merely achieving interventional fairness. As we demonstrate in Section 4.4, it is adept at encompassing all fairness notions that are defined based on interventions. This comprehensive capability significantly enhances the method's applicability in diverse scenarios requiring different fairness considerations.
>
> > W2: Related to the previous point, the paper seems to combine these results in a rather straightforward manner, with limited novel ideas. However, the corresponding theoretical results (e.g., Theorem 4.1) seem sound and no previous work seems to have applied MPDAG identifiability theory to interventional fairness (which, in my opinion, is enough novelty for recommending acceptance, but also justifies why I am hesitant to give a higher score).
>
> Thank you for bringing up this concern and acknowledging the novelty in our work despite these considerations. We would like to clarify that those results are not combined rather straightforward. Once learning an MPDAG from the data with causal discovery algorithms, we propose a modelling technique on the prediction $\hat{Y}$, such that the resulting augmented graph $\mathcal{G^*}$ remains an MPDAG (as detailed in Theorem 4.1), which paves the way to apply the causal identification results for MPDAG on the fairness context. However, the existing causal identification results [Perkovic+; UAI2020] cannot be directly utilized to identify interventional fairness measures on the augmented MPDAG $\mathcal{G^*}$. This is because only the density $f(\mathbf{X}, \mathbf{A})$ over $\mathcal{G}$ is observable, instead of $f(\mathbf{X}, \mathbf{A}, \hat{Y})$ over $\mathcal{G^*}$. The variable $\hat{Y}$ is not directly observable and needs to be learned. To bridge this gap, Proposition 4.3 is proposed formally by exploring the relationship on graph properties and data distribution over $\mathcal{G}$ and $\mathcal{G^*}$. Proposition 4.1 enables the formal identification of the causal effect on the MPDAG $\mathcal{G^*}$ within the context of fairness effectively.

---

> ### Author Response · Authors · 2023-11-20
> **Official Comment by Authors (2/2)**
>
> > W3. Applicability in practice: The proposed method seems to be difficult to employ in practice. First, one has to run a causal discovery algorithm to obtain an MPDAG, and then perform conditional density estimation to obtain the fairness penalty. Furthermore, there is no way of automatically choosing the trade-off parameter (even though this point is not specifically a drawback of the method in this paper).
>
> Thanks for raising this concern. As for the first point, in practical scenarios, fully specifying a causal Directed Acyclic Graph (DAG) often proves challenging due to limited system understanding. Employing a causal discovery algorithm to derive a Maximally Partially Directed Acyclic Graph (MPDAG) is a pragmatic approach in such instances. Regarding the second point, current result in causal effect identification on MPDAGs implies that performing conditional density estimation is an effective strategy presently available. Future research could potentially focus on the development of more efficient model estimation algorithms or the exploration of alternative methods, perhaps by simplifying the model and concentrating on key variables or relationships. Concerning the trade-off parameter, its determination heavily depends the scale of the prediction and the level of fairness desired by the decision-makers. We leave this to the specific data and application context in question.
>
>
> > W4. The authors hint at the possibility of extending their approach to other fairness notions. I think the paper would benefit from a more detailed review of related fairness notions in the appendix (e.g., total/direct/indirect effects, path-specific effects, counterfactual fairness), and how the method could (or could not) be extended
>
> Thanks for the nice suggestion. According to your suggestion, we have added **Appendix F** to present more details on this part.
>
>
> > Questions: In the case of non-identification, the authors propose to replace the penalty term with a sum over all possible MPDAGs by directing the relevant edges. Can the authors provide some intuition/experiments on how this affects the prediction performance? I could imagine that the prediction performance could suffer if the objective is penalized with a large sum of MPDAGs.
>
> Thanks for bring up this question. In the case of non-identification, when the objective is penalized with an average over all possible MPDAGs, the prediction is also an average over all possible MPDAGs. We conduct experiments on various unidentifiable MPDAGs and find that the tradeoff between accuracy and fairness still exists in these graphs. Interestingly, compared with the prediction learnt solely on the ground-truth identifiable MPDAG, the prediction learnt on all possible MPDAGs does not necessarily deteriorate when achieving equivalent levels of unfairness. This could be attributed to the fact that they are penalizing different measures on fairness. We have added **Appendix G11** with experiments and Figure 19 to reflect this observation.

---

### Official Review · Reviewer_T67w · 2023-11-01

**Soundness:** 3 good
**Presentation:** 3 good
**Contribution:** 2 fair
**Rating:** 6
**Confidence:** 3

**Summary:**

The authors explore the challenge of causal fair learning with only partial knowledge of the structural causal model. They approach this issue by assuming the causal model is provided in the form of a maximal partial directed acyclic graph (MPDAG). This graph represents a Markov equivalent class of directed acyclic graphs (DAGs) that align with the available knowledge. Using the MPDAG, the authors' algorithm aims to construct a predictor that not only maximizes prediction accuracy but also adheres to interventional fairness, a causal-based definition of fairness. Unlike existing learning algorithms that ensure interventional fairness but fail to manage the accuracy-fairness trade-off, the proposed algorithm introduces a parameter specifically for this purpose. The authors employ key techniques to identify and estimate the intervention distribution of the predicted outcome. This allows them to evaluate the degree of interventional fairness and incorporate it into the optimization problem of the learning algorithm. Experiments using both synthetic and real datasets confirm the algorithm's capability to control the accuracy-fairness trade-off effectively.

**Strengths:**

1. The paper is well-written and easy to follow.

2. Addressing causal-based fairness with only a fragmentary understanding of the causal graph is vital. The primary challenge in enforcing causal-based fairness is constructing an accurate causal graph. This research's ability to guarantee causal-based fairness using an incomplete causal graph widens its relevance to practical scenarios.

3. Incorporating the techniques from Perkovic 2020 into causal fair learning is interesting. As highlighted by the authors, this approach achieves the identification and estimation of the intervention distribution for the predicted label, thus enhancing the control over the accuracy-fairness trade-off.

4. The experimental findings unequivocally validate the proposed algorithm's capability to control the trade-off between accuracy and interventional fairness.

**Weaknesses:**

1. The algorithm presented seems to be a direct adaptation of the findings from Perkovic 2020. If the intervention distribution over observable endogenous variables can be identified and estimated, it stands to reason that the intervention distribution of the predicted label is also identifable and estimable, given that this label is derived using a known function from the intervened observable endogenous variables. This raises questions about the method's novelity.

2. The approach necessitates the creation and estimation of the generative model for non-admissible attributes. This forms a significant impediment to its application in real-world contexts, especially when the direct causal functions of the partial causal model may not be readily observable.

**Questions:**

1. Could the authors shed light on the contributions and advancements made beyond the scope of Perkovic 2020?

---

> ### Author Response · Authors · 2023-11-19
> **Official Comment by Authors (1/2)**
>
> Thank you for your insightful comments. We greatly appreciate your thoughtful input and address your comments point by point as follows.
>
> > W1: The algorithm presented seems to be a direct adaptation of the findings from Perkovic 2020. If the intervention distribution over observable endogenous variables can be identified and estimated, it stands to reason that the intervention distribution of the predicted label is also identifiable and estimable, given that this label is derived using a known function from the intervened observable endogenous variables. This raises questions about the method's novelity.
>
> Thanks for raising this concern. Under our modeling technique, the interventional distribution of $\hat{Y}$ is identifiable if the interventional distribution over observable endogenous variables is identifiable, as you pointed out. That is because on the MPDAG $\mathcal{G^*}$, given other observables $\mathbf{V'}$, the probability of $\hat{Y}=\hat{y}$ under an intervention on the sensitive attribute $\mathbf{S}$ reduces to conditioning on $\mathbf{S}$, that is $f(\hat{y}|\mathbf{v'},do({\mathbf{s}}))=f(\hat{y}|\mathbf{v'},\mathbf{s})$, thus leading to $f(\hat{y}|do({\mathbf{s}}))=\int f(\hat{y}|\mathbf{v'},\mathbf{s})f(\mathbf{v'}|do(\mathbf{s})) d \mathbf{v'}$, as in Eq.(3) in Proposition 4.1. This reduction is possible because there are no unblocked backdoor paths from $\mathbf{S}$ to $\hat{Y}$ given $\mathbf{V'}$ on $\mathcal{G}^*$. However, this formal result is constructed on a valid causal graph, that is the augmented MPDAG $\mathcal{G^*}$ in our work. Before Proposition 4.1, we do require Theorem 4.1 to claim that the graph $\mathcal{G^*}$ is a valid MPDAG, such that any causal independence tests or causal inference tasks is able to be performed on $\mathcal{G^*}$. It is important to emphasize that Theorem 4.1 is of independent interest. That is because as a general modeling method applicable to any problem, Theorem 4.1 enables various causal inference tasks beyond merely identifying causal effects within the augmented MPDAGs formally.
> Besides, we are unable to directly adopt Perkovic's causal identification findings, because only the density $f(\mathbf{X},\mathbf{A})$ over $\mathcal{G}$ is observable, not $f(\mathbf{X},\mathbf{A},\hat{Y})$ over $\mathcal{G^*}$. The variable $\hat{Y}$ is not directly observable and needs to be learned. To bridge this gap, we introduce Proposition 4.1, which explores the connection between the graph properties and the data distribution across $\mathcal{G}$ and $\mathcal{G^*}$. Proposition 4.1 enables the formal identification of the causal effect on the MPDAG $\mathcal{G*}$ within the context of fairness.
>
>
> > W2: The approach necessitates the creation and estimation of the generative model for non-admissible attributes. This forms a significant impediment to its application in real-world contexts, especially when the direct causal functions of the partial causal model may not be readily observable.
>
> Thanks for raising this concern. Current result in causal effect identification on MPDAGs implies that performing conditional density estimation is an effective strategy presently available. Future research can focus on developing more efficient algorithms for model estimation or exploring alternative approaches that simplifying the model by focusing on key variables or relationships. Furthermore, the practicality of our approach will increase with advancements in causal discovery and causal effect estimation techniques.

---

> ### Author Response · Authors · 2023-11-19
> **Official Comment by Authors (2/2)**
>
> > Questions: Could the authors shed light on the contributions and advancements made beyond the scope of Perkovic 2020?
>
> Thanks for the question. Our principal contribution lies in dealing with fair machine learning with unknown causal graphs. By modeling the predicted outcome $\hat{Y}$ as a function of observable variables and exploring the identifiability results from Perkovic 2020, we are able to identify and estimate the degree of unfairness, allowing us to manage the trade-off between accuracy and fairness. It is important to note that the identifiability results in Perkovic 2020 presuppose the input as an MPDAG. Our Theorem 4.1 claims that our modeling approach ensures that the extended graph $\mathcal{G^*}$ qualifies as an MPDAG. This implies that the identification results from Perkovic 2020, which are contingent on MPDAGs, can be applied to $\mathcal{G^*}$. Nevertheless, we are unable to directly adopt Perkovic's causal identification findings, because only the density $f(\mathbf{X},\mathbf{A})$ over $\mathcal{G}$ is observable, not $f(\mathbf{X},\mathbf{A},\hat{Y})$ over $\mathcal{G^*}$. The variable $\hat{Y}$ is not directly observable and needs to be learned. To bridge this gap, we introduce Proposition 4.1, which explores the connection between the graph properties and the data distribution across $\mathcal{G}$ and $\mathcal{G^*}$. Proposition 4.1 enables the formal identification of the causal effect on the MPDAG $\mathcal{G^*}$ within the context of fairness. Besides, we propose an estimation method as well, while Perkovic's focus is on identifiability.

---

### Official Review · Reviewer_93qw · 2023-11-01

**Soundness:** 3 good
**Presentation:** 3 good
**Contribution:** 3 good
**Rating:** 8
**Confidence:** 4

**Summary:**

This paper investigates the problem of supervised learning under additional fairness constraints in partially known graphs. Most existing measures of causal fairness constraints require prior knowledge of the directed causal graphs that encode the underlying causal relationships between variables. This paper attempts to relax this assumption. Some previous work, such as Fair proposed by Zuo et al., can achieve causal fairness but discards a lot of covariate information, which impairs the prediction performance. In this paper, we propose a method called IFair to trade-off unfairness and prediction performance by solving a constrained optimization problem. Meanwhile, the paper discusses in detail how to identify some causal estimators with theoretical guarantees and provides concrete examples to illustrate the propositions. The effectiveness of this approach has been confirmed in tests on both simulated and real datasets.

**Strengths:**

S1: This paper investigates the novel problem of imposing causal fairness constraints for supervised learning in the absence of detailed causal knowledge.

S2: This paper provides a comprehensive discussion of the related work, and it is clearly written and organized.

S3: Fully specifying causal graphs is one of the main challenges in applying causal fairness in practice. The method presented in this paper relaxes the assumption of fully available causal graphs and is highly motivated.

S4: The IFair method does not require input to be a causal DAG, which is hardly obtained in real-world data. Instead, the proposed method accepts a CPDAG or an MPDAG as the input, which is more feasible.

S5: This paper conduct extensive experiments on both synthetic dataset and real-world dataset, which verifies the effectiveness of IFair method.

**Weaknesses:**

W1: Is the proposed approach still valid when Y is not the last node in the topological ordering, or when the sensitive attribute is not the root node?

W2: Is it necessary for each node to have an arrow pointing to Y in G?

W3: Zuo et al. utilize counterfactual fairness by identifying deterministic non-descendants of sensitive attributes on the MPDAG, which does not seem to require a full understanding of causal graphs. Can the authors provide a more detailed analysis to distinguish between these two approaches?

**Questions:**

Please refer to the weakness part for the questions.

***

After rebuttal: Thank you very much to the authors for their answers to our reviews and for improving the paper during the rebuttal period. The modifications bring valuable content. I read also carefully the other reviews and the corresponding answers. My recommendation is "accept, good paper".

---

> ### Author Response · Authors · 2023-11-19
>
> Thank you for your insightful comments. We greatly appreciate your thoughtful input and address your comments point by point as follows.
>
> > W1: Is the proposed approach still valid when Y is not the last node in the topological ordering, or when the sensitive attribute is not the root node?
>
> Thank you for bringing up this question. Yes, our method does not necessarily require $Y$ to be the last node in topological ordering or the sensitive attribute to be the root node. It is important to note that during the causal discovery process, we learn an MPDAG $\mathcal{G}$ solely from the observational data $(\mathbf{X}, \mathbf{A})$, without considering the outcome variable $Y$. Besides, it's worth highlighting that the identification condition of the total causal effect with $A$ being a singleton, as stated in Section 4.4, specifies that $A$ should not be connected to any undirected edges. This condition is distinct from the root node assumption and is more relaxed than the root node assumption.
>
> > W2: Is it necessary for each node to have an arrow pointing to Y in G?
>
> Thanks for raising this concern. To maximize the data utility, we model $\hat{Y}$ as a function of all the variables in $(\mathbf{X}, \mathbf{A})$ as traditional. However, it is important to consider that not every node necessarily needs to have an arrow pointing to $\hat{Y}$. In some cases, certain variables may not have a direct influence on $\hat{Y}$, resulting in parameters over such edges in the function $\hat{Y}=f(\mathbf{X}, A)$ being close to zero.
>
>
> > W3: Zuo et al. utilize counterfactual fairness by identifying deterministic non-descendants of sensitive attributes on the MPDAG, which does not seem to require a full understanding of causal graphs. Can the authors provide a more detailed analysis to distinguish between these two approaches?
>
> Thanks for the question. The approach proposed in Zuo et al. can also be used to achieve interventional fairness, as demonstrated in Section 3.1. This method, which we refer to as 'IFair' in our experiments, relies solely on deterministic non-descendants of the sensitive attributes for learning fair predictions. This method can achieve almost zero unfairness, but it leads to a greater loss of prediction accuracy due to the ignorance of other features. Thus, it serves as a baseline in our comparative analysis. In contrast, our proposed model, '$\epsilon$-IFair', adopts a more inclusive strategy by utilizing all observational variables, irrespective of their causal relationship with the sensitive attribute $\mathbf{A}$, to derive fair predictions. This approach is articulated as a constrained optimization problem in Section 3.2, striking a balance between fairness and accuracy with vary $\lambda$. This differentiation is visually represented in our experimental section's tradeoff plots, where the 'IFair' model is indicated with a rhombus marker and '$\epsilon$-IFair' with many circles. These markers illustrate how our model, '$\epsilon$-IFair', not only achieves comparable levels of fairness to the 'IFair' model but also preserves higher accuracy, showcasing the effectiveness of our approach in managing the fairness-accuracy trade-off.

---

### Author Response · Authors · 2023-11-19
**General response to reviewers**

We appreciate all reviewers' work and friendly comments. We are encouraged that they found our paper to be important and relevant (Reviewer 93qw, Reviewer T67w, Reviewer U24x), novel and interesting (Reviewer 93qw, Reviewer T67w), highly motivated (Reviewer 93qw), technically sound and identifiability guarantees (Reviewer U24x). Moreover, we are grateful that reviewers found our paper is clearly-written, well-organized and easy to follow (Reviewer 93qw, Reviewer T67w). Reviewer 93qw also found that our paper includes a comprehensive discussion of the related work. We also appreciated that all of the reviewers found our paper includes extensive experiments verifying the effectiveness of the proposed method.

We also appreciate reviewers pointing out our weaknesses. We address their comments point by point and try our best to update the manuscript accordingly. Hope our response addresses the reviewers' concerns.

---

### Meta-Review · Area_Chair_vzFE · 2023-12-11

**Metareview:**

This work presents an approach to causally fair ML that applies to the important setting where the underlying causal graph is only partially known. The particular notion of causal fairness addressed is interventional fairness: the paper addresses the identifiability and estimation of the necessary interventional distributions from observational data, and presents an algorithm for trading off between the accuracy of the predictions and the interventional fairness.

The key strength of the approach given is that it avoids the assumption that one has access to a fully specified causal graph; instead one can work with equivalence classes of DAGS, which you are more likely to have in practice from domain knowledge and observation data. Experiments on real and synthetic datasets show effectiveness of the proposed method in controlling the trade-off between accuracy and interventional fairness. The main weakness of the method is that it requires conditional density estimation to obtain the fairness penalty. The paper would be strengthened by a discussion of the applicability of this approach to other causal notions of fairness.

Acceptance is recommended, as the paper presents a theoretically sound method for causally fair ML that requires only access to an incomplete causal graph; this greatly widens the range of practical scenarios under which causal fairness can be employed.

**Justification For Why Not Higher Score:**

The approach requires conditional density estimation, which can be infeasible.

**Justification For Why Not Lower Score:**

The paper makes a solid technical contribution by providing a principled approach to achieving interventional fairness when the underlying causal graph is only partially known.

---

### Decision · Program_Chairs · 2024-01-16

Accept (poster)